



# Molecular composition and volatility of multi-generation products formed from isoprene oxidation by nitrate radical

Rongrong Wu[1,2], Luc Vereecken[1], Epameinondas Tsiligiannis[3], Sungah Kang[1], Sascha R. Albrecht[1,a], Luisa Hantschke[1], Defeng Zhao[4], Anna Novelli[1], Hendrik Fuchs[1], Ralf Tillmann[1], Thorsten Hohaus[1], Philip T.M. Carlsson[1], Justin Shenolikar[5], François Bernard[6], John N. Crowley[5], Juliane L. Fry[7], Bellamy Brownwood[7], Joel A. Thornton[8], Steven S. Brown[9,10], Astrid Kiendler-Scharr[1], Andreas Wahner[1], Mattias Hallquist[3], Thomas F. Mentel[1*]

[1]Institute of Energy and Climate Research, Troposphere (IEK-8), Forschungszentrum Jülich GmbH, 52428 Jülich, Germany
[2]College of Environmental Sciences and Engineering, State Key Joint Laboratory of Environmental Simulation and Pollution Control, Peking University, 100871, Beijing, China
[3]Department of Chemistry and Molecular Biology, University of Gothenburg, 41296, Gothenburg, Sweden
[4]Department of Atmospheric and Oceanic Sciences & Institute of Atmospheric Sciences, Fudan University, 200438, Shanghai, China
[5]Atmospheric Chemistry Department, Max Planck Institut für Chemie, 55128 Mainz, Germany
[6]Institut de Combustion, Aérothermique, Réactivité et Environnement (ICARE), UPR CNRS, 45071 Orléans, France
[7]Department of Chemistry, Reed College, Portland, OR 97202, USA
[8]Department of Atmospheric Sciences, University of Washington, Seattle, WA 98195, USA
[9]NOAA Chemical Sciences Laboratory, Boulder, CO 80305, USA
[10]Department of Chemistry, University of Colorado, Boulder, CO 80309, USA
[a]present address: SOLIDpower GmbH, 52525 Heinsberg, Germany

[*]*Correspondence to*: Thomas F. Mentel (t.mentel@fz-juelich.de)

**Abstract**

Isoprene oxidation by nitrate radical ($NO_3$) is a potentially important source of secondary organic aerosol (SOA). It is suggested that the second or later-generation products are the more substantial contributors to SOA. However, there are few studies investigating the multi-generation chemistry of isoprene-$NO_3$ reaction, and information about the volatility of different isoprene nitrates, which is essential to evaluate their potential to form SOA and determine their atmospheric fate, is rare. In this work, we studied the reaction between isoprene and $NO_3$ in the SAPHIR chamber (Jülich) under near atmospheric conditions. Various oxidation products were measured by a high-resolution time-of-flight chemical ionization mass spectrometer using $Br^-$ as the reagent ion. They are grouped into monomers ($C_4$- and $C_5$-products), and dimers ($C_{10}$-products) with 1–3 nitrate groups according to their chemical composition. Most of the observed products match expected termination products observed in previous studies, but some compounds such as monomers and dimers with three nitrogen atoms were rarely reported in the literature as gas-phase products from isoprene oxidation by $NO_3$. Possible formation mechanisms for these compounds are proposed. The multi-generation chemistry of isoprene and $NO_3$ is characterized by taking advantages of the time behavior of different products. In addition, the vapor pressures of diverse isoprene nitrates are calculated by different parametrization methods. An estimation of the vapor pressure is also derived from their condensation behavior. According to our results, isoprene monomers belong to intermediate volatility or semi-volatile organic compounds and thus have little effect on SOA formation. In contrast, the dimers are expected to have low or extremely low volatility, indicating that they are potentially



substantial contributors to SOA. However, the monomers constitute 80% of the total explained signals on
average, while the dimers contribute less than 2%, suggesting that the contribution of isoprene $NO_3$ oxidation to
SOA by condensation should be low under atmospheric conditions. We expect a SOA mass yield of about 5 %
from the wall loss and dilution corrected mass concentrations, assuming that all of the isoprene dimers in the
low- or extremely low-volatility organic compound (LVOC or ELVOC) range will condense completely.





## 1. Introduction

Atmospheric submicron aerosols have an adverse effect on air quality, human health and climate (Jimenez et al., 2009; Pöschl, 2005). Secondary organic aerosol (SOA), which is formed from oxidation of volatile organic compounds (VOC) followed by gas-to-particle partitioning, comprise a large fraction (20-90%) of the submicron aerosol mass (Jimenez et al., 2009; Zhang et al., 2007). It is confirmed that a significant proportion of SOA arises from biogenic VOC (BVOC) oxidation (Hallquist et al., 2009; Spracklen et al., 2011).

Isoprene is globally the most abundant non-methane volatile organic compound originating from vegetation, with emissions estimated to be 440-660 Tg yr$^{-1}$(Guenther et al., 2012). Due to its high abundance, as well as its high reactivity with atmospheric oxidants, isoprene plays a significant role in tropospheric chemistry, and its chemistry affects the global aerosol burden and distribution (Carlton et al., 2009; Fry et al., 2018; Ng et al., 2008, 2017; Surratt et al., 2010), although its SOA yield is much lower than those of monoterpenes and sesquiterpenes (Friedman and Farmer, 2018; Kim et al., 2015; Marais et al., 2016; , McFiggans, et al. 2019; Mutzel et al., 2016; Ng et al., 2007, 2008; Surratt et al., 2010; Thornton et al., 2020). Recent model simulations suggested the isoprene-derived SOA production is 56.7 Tg C yr$^{-1}$, contributing up to 41% of global SOA (Stadtler et al., 2018). Observations in southeastern United States suggested that isoprene-derived SOA makes up 17- 48% of total organic aerosol (Hu et al., 2015; Kim et al., 2015; Marais et al., 2016). As a consequence, it is essential to fully characterize the potential of isoprene to form condensable products and its contribution to SOA formation (Carlton et al., 2009).

Generally, isoprene is primarily oxidized by the hydroxyl radical (OH) and somewhat by ozone (O$_3$) in the daytime. At night when the concentration of OH is negligible, the nitrate radical (NO$_3$) and O$_3$ become the predominant oxidants of isoprene. Reaction of isoprene with NO$_3$ is competitive to that with O$_3$ because of its much larger rate constant ($k_{NO_3}$ = 6.5 ×10$^{-13}$ cm$^3$ molecules$^{-1}$s$^{-1}$ and $k_{O_3}$ = 1.28 ×10$^{-17}$ cm$^3$ molecules$^{-1}$s$^{-1}$ at 298 K, respectively, IUPAC), even if the mixing ration of NO$_3$ is 10,000 time lower than that of O$_3$. Although reaction with NO$_3$ only represents ~ 5-6% of isoprene loss, it accounts for a large proportion of isoprene nitrates (~ 40-50%) (Wennberg et al., 2018). Therefore, reaction of isoprene with NO$_3$ is a potential source of SOA. In addition, it is found from both laboratory and chamber experiments that the SOA yield of isoprene from NO$_3$ oxidation is higher than that from OH or O$_3$ oxidation, which is typically less than 5% (Carlton et al., 2009; Dommen et al., 2009; Kleindienst et al., 2007; Kroll et al., 2006). For example, Ng et al. (2008) concluded the isoprene SOA yield from NO$_3$ was in the range of 4.3% to 23.8%, depending on RO$_2$ fate (higher SOA yield when the experiments were dominated by RO$_2$+RO$_2$ rather than RO$_2$+NO$_3$ reaction). Rollins et al. (2009) also observed a high SOA yield from isoprene (14%) when both of its double bonds were oxidized by NO$_3$. In an aircraft study in the southeastern United States, Fry et al. (2018) derived an isoprene-NO$_3$ SOA yield as large as 27% on average under high NO$_x$ conditions, although their mass yield estimation was indirect, and based on a molar yield determination of 9 ± 5%. In light of the relatively high SOA yield from NO$_3$ oxidation, even though only a minor fraction of isoprene is oxidized by NO$_3$, the SOA formed at nighttime would still probably be comparable to that produced at daytime (Brown et al., 2009; Fry et al., 2018).

However, isoprene-NO$_3$ chemistry has received less attention than the extensively studied OH- or O$_3$-initiated oxidation (Barber et al., 2018; Novelli et al., 2020; Peeters et al., 2014; Wang et al., 2018; Wennberg et al., 2018; Whalley et al., 2012). It has been recognized that later-generation oxidation of isoprene by NO$_3$ makes more significant contribution to SOA formation (Carlton et al., 2009; Fry et al., 2018; Rollins et al., 2009).





Nevertheless, although the importance of multi-generation NO$_3$ oxidation of isoprene to SOA formation has
been recognized, few studies extended the investigation beyond the first-generation oxidation, and details of
isoprene-NO$_3$ multi-generation chemistry are still lacking.
Organic compounds, especially highly oxygenated organic molecules (HOM) that have low or extremely
low volatility, contribute significantly to SOA formation by condensation, or even form new particles (Bianchi
et al., 2019; Ehn et al., 2014; Kirkby et al., 2016, Tröstl et al., 2016). Previous studies have confirmed that low-
volatility products from isoprene-NO$_3$ reaction are the major precursors to SOA (Ng et al., 2008; Rollins et al.,
2009; Schwantes et al., 2019). Here the low-volatility compounds refer to gas phase products that allow
fractions to exist in particle-phase, and may include the groups of organic compounds with intermediate
volatility (IVOC, 300<C$^*$<3×10$^6$ μg m$^{-3}$), semi-volatility (SVOC, 0.3<C$^*$<300 μg m$^{-3}$), low volatility (LVOC,
3×10$^{-5}$<C$^*$<0.3 μg m$^{-3}$) and extremely low volatility (ELVOC, C$^*$<3×10$^{-5}$ μg m$^{-3}$) as proposed by Donahue et al.
(2012). In general, SVOC, LVOC and ELVOC can contribute to the SOA formation (Jimenez et al., 2009). In
order to evaluate the potential of oxygenated products to form SOA, information about their vapor pressures is
essential. However, due to the high degree of functionalization, low or extremely low volatility, as well as
uncertainties in quantification and molecular structures, it is challenging to determine the exact vapor pressure
of highly oxidized products. Detailed information on the volatilities of different generation products is lacking,
which impedes the assessment of their contribution to SOA formation.
In this work, we present the results of chamber experiments on isoprene oxidation by NO$_3$ under near
atmospheric conditions, where NO$_3$ was produced in situ by O$_3$ reaction with NO$_2$. Subsequent characteristics of
multi-generation chemistry of isoprene with NO$_3$ are investigated. By examining the time evolution of various
gas-phase products, we propose possible reaction mechanisms that help to get the possible functionalization of
the products. Saturation vapor pressures of the major gas-phase products observed by HR-ToF-CIMS are
predicted by using different parameterization methods that are widely-used or state-of-the-art in literature. In
addition, we estimate the vapor pressure derived from equilibrium partitioning coefficient according to the
condensation behavior of different products in experiments with and without seed aerosols. Based on these
results, the volatility of the major oxidation products stemming from isoprene-NO$_3$ reaction and their potential
to form SOA are evaluated.
**2. Experimental and methods**
**2.1 Atmospheric simulation chamber SAPHIR**
All the data presented here were measured in the atmospheric simulation chamber SAPHIR (**S**imulation of
**A**tmospheric **PH**otochemical **I**n a large **R**eaction Chamber) at Forschungszentrum Jülich, Germany, which is
designed to investigate the oxidation processes of both biogenic and anthropogenic trace gases and formation of
secondary particles and pollutants under near atmospheric conditions. The SAPHIR chamber is a double-walled
Teflon (FEP) cylinder with a volume of 270 m$^3$ (5 meters in diameter and 18 meters in length). The large
volume-to-surface ratio (1 m) allows experiments to be conducted under natural conditions and reduces
interference from the chamber walls. The chamber is equipped with a shutter system which can be opened to
admit sunlight for photochemical experiments or closed to mimic nighttime conditions. There are two fans
inside the chamber to ensure good mixing of trace gases (within 2 minutes). The chamber is filled with synthetic


air made from mixing of ultrapure nitrogen and oxygen (Linde, purity ≥ 99.99990%) and is slightly over-
pressured (~ 35 Pa) to prevent intrusion of outside air into the chamber. Due to small leakage (~ 7 $m^3$ $h^{-1}$) and
gas consumption by instrument sampling, a replenishment flow is provided by a flow control, which leads to a
dilution rate of 4%–7% per hour. A more detailed description of the chamber set-up and its characterization can
be found elsewhere (Rohrer et al., 2005).

**2.2 Experiment description**

A series of experiments investigating the oxidation of isoprene by $NO_3$ were conducted in the SAPHIR chamber
in August 2018 (ISOPNO$_3$ campaign) under different chemical conditions. In this work, we primarily focus on
an experiment conducted on 08 August 2018 that examined the fast oxidation of isoprene by $NO_3$ (up to ~ 130
pptv) without seed aerosols. The experiment was performed under dry (RH < 5%) and dark condition, and
employed injections of $O_3$ and $NO_2$ as source of $NO_3$, where $O_3$ was generated by a silent discharge ozoniser
(O3onia), and high-purity $NO_2$ was introduced from a gas bottle (Linde, purity >99%).
Before the experiment, the chamber was flushed overnight with a total amount of ~ 1800 $m^3$ synthetic air to
minimize any remaining contamination. At the beginning of the experiment, the chamber air was slightly
humidified (RH< 0.1%) by flushing water vapor from boiling Milli-Q® water into the chamber. Thereafter, $O_3$
and $NO_2$ were added to the chamber in succession, and their concentrations in the chamber after injection were
approximately 100 and 25 ppbv, respectively, as shown in Fig. 1. After that, ~10 ppbv of isoprene was injected
using a GC syringe, initiating the reaction with $NO_3$. The period between the first and second injection is
defined as "step I", as so on for the other three periods. After almost complete consumption of isoprene, another
~100, 30, and 10 ppbv of $O_3$, $NO_2$, and isoprene, respectively, were added. After another ~ 1.5 hours, the
chemistry was accelerated again by the third injection, and the concentrations of $O_3$, $NO_2$, and isoprene reached
~ 100, 25, and 10 ppbv, respectively, after the injection. Two hours later, the fourth addition was made and the
concentrations of $O_3$ and $NO_2$ increased to approximately 115 ppbv and 30 ppbv, respectively, aiming to
promote further oxidation of early generation products. In total the system was kept running for about 7.5 h.
According to the modeling results, approximately 90% of the isoprene reacted with $NO_3$, indicating that reaction
with $O_3$ was a minor sink of isoprene in our system.
A complementary experiment was conducted on 14 August 2018 under similar conditions but with seed
aerosols. Approximately 60 μg $m^{-3}$ of ammonium sulfate aerosol was added at the beginning of the experiment.
Thereafter, approximate 100 and 20 ppbv of $O_3$ and $NO_2$ were introduced to the chamber to produce $NO_3$,
followed by addition of ~10 ppbv of isoprene in about 30 minutes later (see Fig. S1). Another 6 ppbv of $NO_2$
and 10 ppbv of isoprene were added about one hour later to accelerate the reaction. At the last injection, only $O_3$
(~ 50 ppbv) and $NO_2$ (~ 7 ppbv) were added, similar as for the experiment without seeds. The experiment lasted
for about 8 h. The results were used to investigate the condensation behavior of various gas-phase products from
isoprene oxidation, aiming to estimate equilibrium partitioning coefficients and vapor pressures.

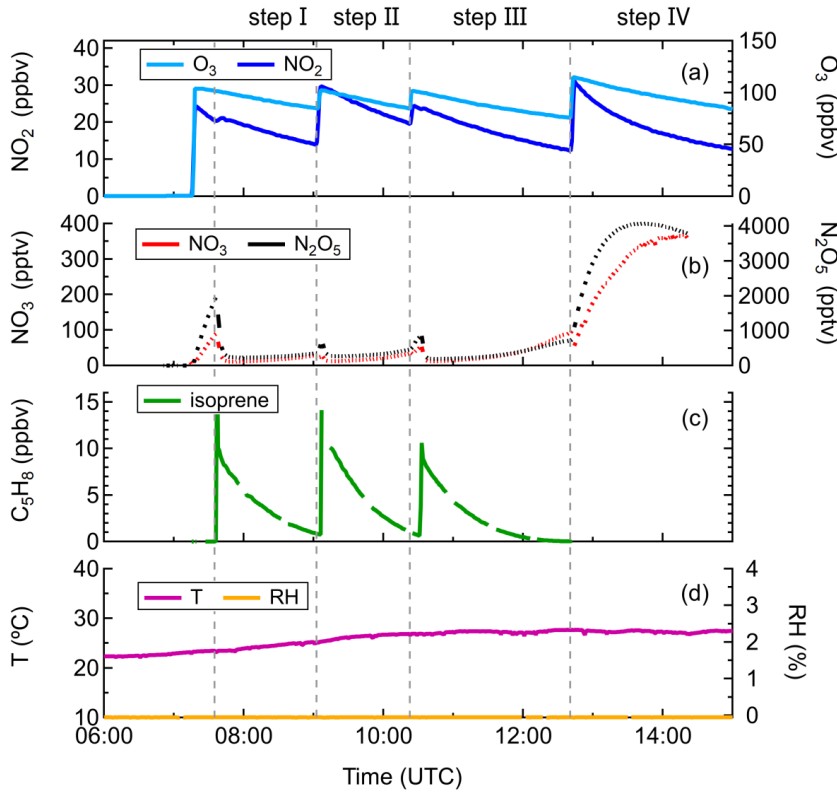

**Figure 1: Measurements of (A) O₃ and NO₂, (B) NO₃ and N₂O₅, (C) isoprene and (D) temperature and relative humidity in the chamber during the experiment on 08 August, 2018.**

### 2.3 Instrumentation

A high-resolution time-of-flight chemical ionization mass spectrometer (HR-ToF-CIMS, Aerodyne Research Inc., hereafter CIMS) was used to continuously measure the gas-phase products from isoprene oxidation by $NO_3$. The ToF-MS was operated in 'V' mode with the mass resolution power between 3000–4000 Th/Th. In order to reduce the losses of $HO_2$ radicals and HOM on the tubing, a customized inlet (Albrecht et al., 2019) was directly connected to the chamber. The CIMS was operated in negative ion mode using $Br^-$ as the reagent ion, which is selective to polar species such as acids, hydroxy or nitroxy carbonyls, as well as $HO_2$ radicals (Albrecht et al., 2019; Ng et al., 2008; Rissanen et al., 2019; Riva et al., 2019).

Bromide ions were generated by passing a mixture of 10 standard cubic centimeters per minute of 0.4% $CF_3Br$ in nitrogen and 2 standard liter per minute nitrogen through a 370 MBq [210]Po source (Type P-2021-5000, NDR Static Control LLC, USA), resulting in ~$10^5$ ion counts per second (Albrecht et al., 2019). In our system, most compounds were detected as adducts with $Br^-$, but some strong acidic compounds like nitric acid were also detected as deprotonated ions. The isotope distribution of [79]Br and [81]Br is approximately 1:1, therefore two signals appear at $m/z$ = MW+79 and $m/z$ = MW+81 with MW being the molecular mass of the molecule that is detected as cluster with $Br^-$. In this work, we will use Thomson (Th) as the unit for mass-to-charge ($m/z$), and the


*m/z* of molecules discussed in following include the mass contribution from Br⁻ *(m/z 79)* if there is no other
annotation.
In order to have an indicator of the CIMS performance, perfluoropentanoic acid (PFPA) was used as an
internal standard. The CIMS was optimized to gain a maximum signal of [HO$_2$*Br]⁻ isotopes, which are weakly
bounded clusters. This was achieved by adjusting step by step the electrostatic field in the transfer stage to
minimize fragmentation. During the campaign, the settings of CIMS were kept unchanged to keep a similar
performance. However, the signal of reagent ion Br⁻ decreased by about 65% (from ~ 100, 000 to 34, 000 counts
s⁻¹) over the campaign duration of four weeks. In order to minimize the effect of drift in performance, we used
the normalized (by the sum of the total ion counts) signals for analysis. The sensitivity for total carbon was
calculated by determining the slope of wall-loss corrected total carbon signals detected by CIMS (only the
identified peaks were considered) versus isoprene consumed. As illustrated in Fig. S2a, the CIMS sensitivities
were roughly identical in two experiments (0.026 ± 0.002 norm. count s⁻¹ ppbv⁻¹ on 08 August, and 0.022 ±
0.001 norm. count s⁻¹ ppbv⁻¹ on 14 August), indicating that different experimental conditions over two days had
an insignificant impact on CIMS sensitivity for total carbon and thus the data from these days are comparable.
In addition, an inter-comparison of measurements by Br⁻ CIMS and I⁻ CIMS were made. As shown in Fig. S2b,
the measurements of C$_5$H$_6$N$_2$O$_8$ from the two instruments are well linearly correlated with each other at the early
oxidation stages. However, the correlation coefficient somewhat changes between the two experiments, which is
possibly related to the interference from isomers and the differences in sensitivity between the two instruments.
In general, the performance of Br⁻ CIMS was stable and the data taken by it are reliable.
The mass spectra data were processed using the software "Tofware" embedded in Igor as provided by
Aerodyne Research Inc. (https://www.tofwerk.com/software/tofware/?cn-reloaded=1). Peaks detected in the
mass spectra could be isolated and identified according to their exact mass, and molecular formulas and the
corresponding intensities were obtained by high-resolution peak fitting. Due to a lack of authentic standards for
the products, it is difficult to quantitatively determine their individual absolute concentrations, but we have
calculated the bulk sensitivity for organonitrates by determining the slope of total organic nitrate signals
detected by Br⁻ CIMS versus the alkyl nitrate concentrations measured by a thermal dissociation cavity ring-
down spectrometer, as shown in Fig. S2c. The estimated bulk sensitivities for organonitrates are 0.016 ± 0.001
and 0.022 ± 0.001 norm. count s⁻¹ ppbv⁻¹on 08 August and 14 August, respectively, comparable to the sensitivity
for total carbon, but smaller than the sensitivity for salicylic acid determined by an independent calibration (163
norm. count μg⁻¹ on average as shown in Fig. S2d, equal to 0.07 norm. count s⁻¹ ppbv⁻¹). The bulk sensitivity for
organonitrates enables estimation of the absolute concentrations of products assuming that they have identical
sensitivity. In this study we use the normalized signals instead of absolute concentrations for analysis. This is
sufficient here because our analysis focuses on the time evolution of signals and the relative changes of
intensities, so the absolute concentrations are not necessarily needed. The sensitivity derived above is only used
to convert the signals of dimers to concentrations in order to estimate the SOA yield.
Isoprene was measured by a Vocus proton transfer reaction time-of-flight mass spectrometer (Aerodyne
Research Inc., hereafter Vocus), which has a higher mass resolving power (nominal 10000 Th/Th) and less inlet
wall losses and sampling delays compared to traditional PTR-MS (Krechmer et al., 2018). The mixing ratio of
O$_3$ was monitored by an UV absorption instrument, and that of NO$_2$ was monitored by a chemiluminescence
instrument and a custom-built cavity ring-down spectrometer (CRDS). The concentrations of NO$_3$ and N$_2$O$_5$



were detected by two custom-built CRDS instruments (Dubé et al., 2006; Sobanski et al., 2016). In addition,
temperature and pressure inside the chamber were monitored by an ultra-sonic anemometer and a pressure
sensor, respectively. The relative humidity was primarily detected as water mixing ratio by a Picarro CRDS
instrument (Crosson, 2008).
The particle number concentrations and their size distributions were measured by a condensation particle
counter (TSI 3783, hereafter CPC) and a scan mobility particle sizer (TSI 3081 electrostatic classifier combined
with TSI 3025 CPC, hereafter SMPS). The aerosol chemical composition was identified by a high-resolution
time of flight aerosol mass spectrometer (HR-ToF-AMS, Aerodyne Research Inc., hereafter AMS). The
ionization efficiency of AMS was determined by using the monodisperse aerosol generated from $NH_4NO_3$ and
$(NH_4)_2SO_4$ solutions. The collection efficiency (CE) could be estimated based on the particle mass concentration
yielded from AMS and that derived from SMPS. In this study, the average CE value of 0.5 is used for correction.
**2.4 Methods to estimate saturation vapor pressure**
The pure liquid saturation vapor pressure is a thermodynamic metric relevant for the partitioning equilibrium of
organic molecules, which determines their propensity to form SOA (Compernolle et al., 2011; O'Meara et al.,
2014; Pankow and Asher, 2008). Due to their complex functionalities and low or extremely low volatility, it is
challenging to determine the vapor pressures of highly oxidized molecules. As a result, theoretical and
semiempirical methods are usually used for vapor pressure estimation. Commonly used semiempirical methods
include composition-activity (CA), group-contribution (GC), and structure-activity (SA) methods. The CA
methods are the easiest to use, as they only require information on molecular composition for estimation. They
are widely applied in context of the two-dimensional volatility basis set (2D-VBS) (Donahue et al., 2011). For
GC methods, the exact functional groups are required to calculate the saturation vapor pressure. The SIMPOL.1
(Pankow and Asher, 2008), the parameterization as described by Nannoolal et al. (2008), and EVAPORATION
(Compernolle et al., 2011) are three widely used GC methods. Structure-activity methods can provide more
accurate estimates with sophisticated treatments of intramolecular interactions like intramolecular hydrogen-
bonding (Bilde et al., 2015). However, detailed molecular properties such as boiling point and evaporation
enthalpy are required for estimation, which are generally obtained by complex and time-consuming quantum
chemical calculations. Therefore, SA methods are not applied for vapor pressure estimation in this study.
Saturation concentration ($C^*$, mass based) is related to saturation vapor pressure and can be calculated
following Eq. (1) (Donahue et al., 2006). The $\log_{10}(C^*)$ is a metric used in the 2D-VBS method to evaluate the
volatility of organic molecules.
$$C_i^* = \frac{M_i 10^6 \zeta_i p_i^\circ}{RT} \qquad\qquad (1)$$
where $R$ ($8.206 \times 10^{-5}$ $m^3$ atm $K^{-1}$ $mol^{-1}$) is the gas constant, $T$ (K) is the temperature, $M_i$ (g $mol^{-1}$) is the molecular
weight of compound $i$, $\zeta_i$ is the activity coefficient of compound $i$ and here is assumed to be 1 (Donahue et al.,
2006), $p_i^\circ$ (atm) is the pure liquid saturation vapor pressure at temperature $T$ (298 K).
In this study, different CA methods are applied to calculate the saturation vapor pressures of various
oxidation products from isoprene reaction with $NO_3$. These include parameterizations that were constrained by
chamber measurements as proposed by Donahue et al. (2011), Mohr et al. (2019), and Peräkylä et al. (2019).
Further we test the GC methods proposed by Nannoolal et al. (2008), Pankow and Asher (2008, SIMPOL.1),





and Compernolle et al. (2011, EVAPORATION). All the methods used in this study are summarized in Table 1.
The calculations of EVAPORATION and the Nannool method were done via the online molecular and
multiphase property prediction facility UManSysProp
(http://umansysprop.seaes.manchester.ac.uk/tool/vapour_pressure). For the latter the boiling point
parameterization method needs to be predefined, and that from Nannoolal et al. (2004) was adopted as
recommended by O'Meara et al. (2014). The information about molecular structures needed for the calculation
is inferred from mechanistic information, which is described in detail in Sect. 2.5.
**Table 1: Summary of estimation methods of saturation vapor pressure used in this study**

| Estimation method | Methodology | Input information | | | Reference |
| --- | --- | --- | --- | --- | --- |
| | | molecular formula | functional groups | others | |
| Donahue et al. | CA[a] | √ | | | Donahue et al., 2011 |
| Mohr et al. | CA | √ | | | Mohr et al., 2019 |
| Peräkylä et al. | CA | √ | | | Peräkylä et al., 2020 |
| Nannoolal et al. | GC[b] | √ | √ | √ [d] | Nannoolal et al., 2008 |
| SIMPOL.1 | GC | √ | √ | | Pankow and Asher, 2008 |
| EVAPORATION | GC | √ | √ | | Compernolle et al., 2011 |
| This study | EXP[c] | | | | |

[a] abbreviation of composition-activity method; [b] abbreviation of group-contribution method; c abbreviation of
experimental method; [d] boiling point parameterization method is also required to be defined.
In addition, we take advantage of the measurements in this study to calculate the gas-particle equilibrium
partitioning coefficient (K) by comparing experiments with and without seed aerosols. The partitioning
coefficient K can be converted to saturation concentration C* by Eq. (2).
$$K_i = \frac{C_{i,p}}{C_{i,g} \times C_{OA}} = \frac{1}{C_i^*} \qquad (2)$$
where $C_{i,g}$ and $C_{i,p}$ are the gas- and particle-phase concentrations (μg m$^{-3}$) of species i, respectively, and $C_{OA}$ is
the organic aerosol concentration (μg m$^{-3}$). In this study, $C_{i,g}$ is signal of species $i$ from CIMS in the experiment
with seeds, and $C_{i,p}$ is the difference of signals between experiment without and with seeds (under the same
isoprene consumption condition). The $C_{OA}$ in the experiment with seeds is in a range of 1-4 μg m$^{-3}$.
**2.5 Pathways to the multifunctional oxidation products**
**2.5.1 Basic peroxy and alkoxy radical chemistry**
As mentioned before, information about molecular structures (at least functional groups) is required to calculate
vapor pressures by using GC methods. Although the high-resolution ToF-CIMS allows for determining
chemical composition of the detected ions, it is unable to provide information about molecular structures, so that
the constitutional or configurational isomers with the same mass cannot be distinguished without additional





information. Fortunately, knowledge of detailed chemical formation mechanisms can help inferring the
molecular structure information. However, the development of a comprehensive, multi-generational kinetic
mechanism for $NO_3$-initiated oxidation of isoprene is outside the scope of the current paper. Instead, in order to
link the observed mass peaks to representative molecular structures, we developed a framework tracing the
chemical oxidation mechanisms by taking well-known oxidation steps to predict the most likely isomeric forms
of the functionalized products formed in the isoprene oxidation. For this purpose, we rely on the extensive
literature on isoprene, alkylperoxy radical, and alkoxy radical chemistry (Atkinson, 2007; Atkinson and Arey,
2003; Bianchi et al., 2019; Crounse et al., 2013; Ehn et al., 2014; Jenkin et al., 2015; Kwan et al., 2012; Mentel
et al., 2015; Ng et al., 2008; Orlando et al., 2003; Orlando and Tyndall, 2012; Rollins et al., 2009; Schwantes et
al., 2015; Vereecken and Francsico, 2012; Vereecken and Peeters, 2010; Wennberg et al., 2018; Ziemman and
Atkinson, 2012). This framework is depicted in the supporting information and will be discussed in more detail
in Sect. 2.5.2 and Sect. 2.5.3. They are based on the following main reactivity trends.
Generally, $RO_2$ radicals can react with other $RO_2$ and $HO_2$ radicals. There are three major channels for the
reaction between two $RO_2$ radicals, leading to alkoxy radicals (RO) (Reaction R1a), as well as termination
products like alcohols, aldehydes or ketones (Reaction R1b) and accretion products (Reaction R1c). These
reactions should take place with the first-generation peroxy radicals, as well as with the higher generation $RO_2$
radicals formed in the later oxidation steps. Hydroperoxides can be formed from the reaction of $RO_2$ with $HO_2$
radicals (Reaction R2a). This reaction can also yield alkoxy radicals (Reaction R2b).
$RO_2\cdot + R'O_2\cdot \longrightarrow RO\cdot + R'O\cdot + O_2$ (R1a)
$RO_2\cdot + R'O_2\cdot \longrightarrow ROH + R'_{-H}{=}O + O_2$ (R1b)
$RO_2\cdot + R'O_2\cdot \longrightarrow ROOR' + O_2$ (R1c)
$RO_2\cdot + HO_2\cdot \longrightarrow ROOH + O_2$ (R2a)
$RO_2\cdot + HO_2\cdot \longrightarrow RO\cdot + \cdot OH + O_2$ (R2b)
In the presence of $NO_x$, $RO_2$ radicals can also react with NO and $NO_2$, leading to the formation of alkoxy
radicals (R3a), organic nitrates (R3b), and peroxynitrates (R4) (including peroxyacyl nitrates, PANs, if R =
$R'C(O)$–). The channel that results in RO radicals is the major pathway for the reaction of $RO_2$ radicals with NO
(Ziemann and Atkinson, 2012). However, reactions of $RO_2$ radicals with NO (Reaction R3a and R3b) can be
neglected in this study due to the high $O_3$ concentration, which results in rapid conversion of NO to $NO_2$. The
peroxynitrates formed from the reaction of $RO_2$ with $NO_2$ will undergo rapid thermal decomposition under our
experimental conditions, with exception of PANs. The reaction of $RO_2$ with $NO_3$ radicals mainly forms $NO_2$ and
alkoxy radicals (Reaction R5), which will continue the radical chains (Reaction R7).
$RO_2\cdot + NO \longrightarrow RO\cdot + NO_2$ (R3a)
$RO_2\cdot + NO \longrightarrow RONO_2$ (R3b)
$RO_2\cdot + NO_2 + M \leftrightarrow ROONO_2 + M$ (R4)
$RO_2\cdot + NO_3 \longrightarrow RO\cdot + NO_2 + O_2$ (R5)
In addition to biomolecular reactions, intramolecular rearrangement (H-migration) is a competitive reaction
pathway for $RO_2$ radicals. $RO_2$ radicals can undergo H-migration to form a hydroperoxy functionality (–OOH)
and a radical site that can subsequently recombine with an $O_2$ molecule, leading to the formation of a new, more
oxidized substituted $RO_2$ (Reaction R6). This process is the so-called "autoxidation" path and has been





confirmed as a significantly important way for SOA formation (Crounse et al., 2013; Ehn et al., 2014; Mentel et
al., 2015; Praske et al., 2018; Rissanen et al., 2014). The rates of $RO_2$ H-migration are strongly dependent on the
structure of $RO_2$ radicals, and the most likely routes can be derived based on the structure-activity relationship
proposed by Vereecken and Nozière (2020).
$RO_2\cdot \; \rightarrow \; HOOQ\cdot \; ; \; HOOQ\cdot + O_2 \; \rightarrow \; Q(OOH)O_2\cdot$        (R6)

The RO radicals formed in in the reaction of $RO_2 + RO_2$ typically have three accessible pathways,

including isomerization by H-migration (Reaction R7a), fragmentation (Reaction R7b) and less important here,
reaction with $O_2$ (Reaction R7c). Like H-migration in $RO_2$, rearrangement by H-shift in RO radicals leads to the
formation of more oxidized $RO_2$ radicals. Fragmentation leads to smaller carbon chains, and this becomes more
important for alkoxy radicals with a higher number of (oxygen-bearing) substituents (Vereecken and Peeters,

2009, 2010).

$RO\cdot \; \rightarrow \; HOQ\cdot \; ; \; HOQ\cdot + O_2 \; \rightarrow \; R(OH)O_2\cdot$        (R7a)
$RO\cdot \; \rightarrow \; R^{'}{=}O + R^{''}\cdot$        (R7b)
$RO\cdot + O_2 \; \rightarrow \; R{=}O + HO_2\cdot$        (R7c)

In addition to the above general reaction pathways, we include a number of other reactions in the

framework, such as fragmentation of peroxy radicals, epoxidation of β-OOH alkyl radicals, and unimolecular
termination of nitrooxy or hydroperoxyl peroxy radicals. Details can be found in the supporting information.
**2.5.2 Formation of first-generation products**
Here "first-generation products" refers to the closed-shell compounds from the first attack of $NO_3$ at the
isoprene double bonds, while "second-generation products" follow an addition of $NO_3$ to the remaining double
bond (or any other oxidation reaction) of a first-generation product. Addition of a $NO_3$ radical to one of isoprene
double bonds and subsequent addition of $O_2$ to the resulting (delocalized) radical sites leads to the formation of
nitrooxy alkylperoxy radicals ($INO_2$, $C_5H_8NO_3$). Since isoprene contains two double bonds, $NO_3$ can attack any
of the four positions on the conjugated carbon bonds, resulting in eight possible $INO_2$ isomers (including six
constitutional and two conformational isomers), as shown in Scheme S1. However, both theoretical and
experimental studies suggest that the addition occurs preferably at the primary and terminal carbons, wherein C1
addition seems to be preferred over C4 addition (Schwantes et al., 2015; Suh et al., 2001; Wennberg et al., 2018).
As the GC methods have limited or no ability to distinguish between positional isomers (Kurten et al., 2016), we
take exemplarily the products following the C1 addition for the vapor pressure analysis in this study.

The initial peroxy radicals ($C_5H_8NO_3$) can undergo rearrangement by H shift from C–H bonds with

subsequent $O_2$ addition, yielding new –OOH functionalized peroxy radicals (Reaction R6). Repeating this
process can lead to the formation of a series of peroxy radicals and termination products with stepwise
increasing number of oxygen atoms by 2, as shown in the conceptual scheme Scheme S2. This is the $RO_2$
autoxidation channel and the molecular formula of peroxy radicals formed via consecutive $O_2$ additions can be
represented as $C_5H_8NO_{(3+2n)}$ ($n \geq 1$, number of autoxidation steps). The autoxidation chain can be terminated
when the H-shift occurs at a carbon with an –OOH or –$ONO_2$ group attached, leading to carbonyl formation
with OH or $NO_2$ loss (Anglada et al., 2016; Bianchi et al., 2019; Vereecken, 2008; Vereecken et al., 2004). The





closed-shell products formed in these termination steps have the general molecular formula $C_5H_7NO_{(5+2n-1)}$ (OH
loss channel) or $C_5H_8O_{(3+2n-2)}$ (NO$_2$ loss channel).

The $C_5H_8NO_{(3+2n)}$ peroxy radicals can also react with HO$_2$ radicals to form –OOH functionalized

termination products with the general molecular formula $C_5H_9NO_{(3+2n)}$ (Reaction R2a), or yielding the alkoxy
radicals $C_5H_8NO_{(3+2n-1)}$ (Reaction R2b). In addition, the $C_5H_8NO_{(3+2n)}$ peroxy radicals can react with other RO$_2^{\cdot}$
radicals (Reaction R1a-R1c). The reaction R1a leads to the formation of alkoxy radicals ($C_5H_8NO_{(3+2n-1)}$) while
R1b forms closed-shell products either with a carbonyl group ($C_5H_7NO_{(3+2n-1)}$) or a hydroxyl group
($C_5H_9NO_{(5+2n-1)}$). Alternatively, dimers can be formed following Reaction R1c, which have then two –ONO$_2$
groups and at least 8 oxygen atoms depending on the formula of RO$_2$ radicals involved, as shown in Table S1.

The alkoxy radicals from reactions R1a and R2b can undergo unimolecular rearrangement by H shift with

subsequent O$_2$ addition, similar to the RO$_2$ radicals, forming new RO$_2$ radicals with a –OH group (Reaction
R7a). As mentioned above, when the H-shift occurs at a carbon with an –OOH or –ONO$_2$ group attached, the
resulting intermediates tend to lose an OH group or NO$_2$ (Bianchi et al., 2019), yielding the closed-shell
carbonyl products with general formulas $C_5H_7NO_{(5+2n-2)}$ or $C_5H_8O_{(3+2n-3)}$ respectively, as shown in the conceptual
scheme Scheme S3. The newly-formed RO$_2$ radicals from alkoxy H-shift channel can follow the peroxy
pathways (Reaction R1-R6) like other RO$_2$ radicals, leading to a diversity of compounds like hydroperoxides
(Reaction R2a, $C_5H_9NO_{(3+2n+1)}$), alcohols (Reaction R1b, $C_5H_9NO_{(3+2n)}$), aldehydes (Reaction R1b, $C_5H_7NO_{(3+2n)}$)
as well as accretion products (Reaction R1c, $C_{10}H_{16}N_2O_x$), as depicted in Scheme S3. Alternatively, they can
also yield alkoxy radicals again following reactions R1a and R2b and continue so on. Furthermore, the alkoxy
radicals can break apart into two fragments according to Reaction R7b.

In general, the alkoxy reaction pathways diversify the parity of the oxygen number of the products from the

reaction of isoprene with NO$_3$, and the compounds formed via these reactions generally have one less or one
more oxygen atom compared to those formed from straight peroxy reaction pathways. With help of the
mechanistic framework described above, we can infer the functionality of first-generation products. This is
exemplified in Scheme S5 and S6 for the major first-generation C$_5$ products. In addition, the reaction pathways
and their corresponding structures of the first-generation C$_{10}$ dimers ($C_{10}H_{16}N_2O_x$) are summarized in Scheme
S13.

### 383    2.5.3 Formation of second-generation products

Nitrate radicals can oxidize the first-generation products once again at the double bond remaining ($k_{NO_3}$(298K) ~
3-11×10$^{14}$ cm$^3$ molecule$^{-1}$ s$^{-1}$, Wennberg et al., 2018). This leads eventually to "second-generation" products that
contain at least two nitrogen atoms. Addition of NO$_3$ radical to the remaining double bond of the first-generation
products results in the formation of dinitrooxy peroxy radicals. We assume that dinitrooxy peroxy radicals can
undergo unimolecular and bimolecular reactions (Reaction R1–R6) in analogy to nitrooxy peroxy radicals,
which lead to secondary products containing two or more nitrogen atoms, as summarized in the conceptual
scheme Scheme S4.

The reaction of first-generation nitrooxy peroxy radicals with NO$_2$ can also yield 2N-compounds (Reaction

R4), however these 2N-compounds ought to be under first-generation products by definition. Such species are
not discussed in detail here but will be covered to catch the diversity of the functionalities for the vapor pressure
estimation. With the help of this secondary reaction framework, we can propose functional groups for the major





second-generation products. Scheme S8 – S10 depict the detailed (possible) reaction pathways that lead to the
formation of detected $C_5$ dinitrates, as well as their possible structures. Furthermore, the proposed formation
mechanism and their structures for $C_5$ trinitrates are shown in Scheme S12, while those for the second-
generation $C_{10}$ dimers ($C_{10}H_{17}N_3O_x$ and $C_{10}H_{18}N_4O_x$) are depicted in Scheme S13.

### 2.5.4 Formation of fragmentation products

In addition to the multigenerational $C_5$ and $C_{10}$ products, fragmentation products can be formed from the
reaction of isoprene with $NO_3$. As mentioned above, the alkoxy radicals can undergo C–C bond scission,
producing a carbonyl compound and an alkyl fragment (Reaction R7b). As shown in Scheme S7, when the
secondary nitrooxy alkoxy radicals from the further oxidation of $C_5$ carbonyl compounds ($C_5H_8O_2$ and $C_5H_8O_3$
here) undergo unimolecular decomposition, $C_4$ carbonyl products ($C_4H_7NO_5$ and $C_4H_7NO_6$, respectively) are
formed as well as formyl radicals. Since the bond fission can occur at different positions, the generation of more
reactive $C_2$ and $C_3$ carbonyl compounds are possible. In addition, the $C_4$ carbonyl compounds are possibly
generated through peroxy radical arrangement by 1,4 H-shift and subsequent acyl radical bond scission reactions
(see Scheme S7). The $C_4$ dinitrates can be formed following similar chemistry, as depicted in Scheme S11.

### 2.5.5 Candidate structures for vapor pressure estimation

Among all gas-phase products detected by CIMS, we selected 32 major representative organonitrates formed
from isoprene oxidation by $NO_3$ radicals. Their structures are rationalized by the corresponding molecular
formulas and proposed formation mechanisms in the reaction framework. Table S2 summarizes all the
exemplified structures used for vapor pressure estimation. The functional groups covered in the selected
structures include nitrate, hydroxyl, ketone, aldehyde, carboxylic acid, peroxide, hydroperoxide, hydroperoxy
acid, peroxynitrate, peroxyacyl nitrate and epoxide. The structural information allows calculation of the
saturation vapor pressure by GC methods.

### 3. Results and discussion

### 3.1 Chemical composition of oxidation products

Figure 2 illustrates the average mass spectra of the whole experiment measured by Br⁻CIMS for isoprene-$NO_3$
reaction. Chemical sum formulas were attributed to most of the detected ions. The gas-phase products were
separated into two major groups according to their chemical composition, including monomers comprising $C_5$
compounds and dimers containing $C_{10}$ compounds. There were also products from decomposition reactions with
$C_{<5}$, which were merged into monomers. The monomers and dimers were further classified into five subgroups
as follows. Monomers consisting of compounds with one nitrogen atom (hereafter 1N-monomers) and two or
three N atoms (2N- or 3N-monomers) mainly accumulate in $m/z$ 220–280 Th, $m/z$ 300–340 Th and 350–390 Th,
respectively, while dimers containing compounds with two N atoms (2N-dimers) and three N atoms (3N-dimers)
appear in $m/z$ 370–440 Th and 450–520 Th, respectively. As shown in Fig. 2, the signal intensities decrease
from 1N-monomers, 2N-monomers, 2N-dimers to 3N-monomers and 3N-dimers. Many of the compounds
detected in this work were also observed in previous isoprene-$NO_3$ systems (Kwan et al., 2012; Ng et al., 2008;
Schwantes et al., 2015). In this work, only closed-shell products are considered for analysis.

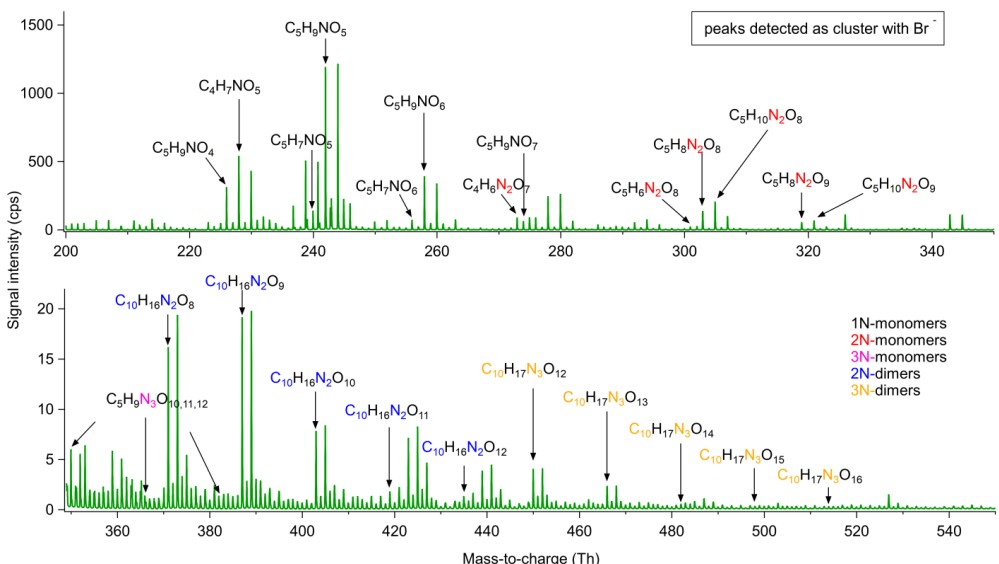

**Figure 2: Averaged mass spectra for isoprene-NO₃ experiment on 8 August, 2018. Molecular formulas were determined according to the accurate mass data provided by HR-ToF-CIMS.**

The 1N-monomer $C_5H_9NO_5$ at $m/z$ 242 is the dominant product formed from the NO₃-induced isoprene oxidation in our experiment, followed by the 1N-decomposition product $C_4H_7NO_5$ at $m/z$ 228. In addition to $C_5H_9NO_5$, several analogues with molecular formulas $C_5H_7NO_{4-7}$ and $C_5H_9NO_4$ are in relatively high abundance. $C_5H_{8,10}N_2O_{8,9}$ and $C_5H_9N_3O_{10-12}$ are the major 2N- and 3N-monomers. Their signal intensities are one to two orders of magnitude lower than those of 1N-monomers. According to the chemical composition, the 1N-monomers are likely to be the first-generation products from NO₃ oxidation of isoprene, while the 2N- and 3N-monomers probably arise from the further oxidation of 1N-monomers by NO₃, which therefore should be second- or later-generation products. As mentioned before, the reaction of nitrooxy alkylperoxy radicals with NO₂ can lead to the formation of peroxynitrates (for the special case peroxyacyl nitrates, PAN-like) containing two N atoms. The peroxynitrates will decompose rapidly under experimental conditions, whereas the PAN-like compounds are more stable (with lifetimes ranging from minutes to weeks at 298K and ambient temperature). Such C₅ PAN-like compounds are isomers of aforementioned 2N-monomers, but ought to be first-generation products. In addition to C₅-2N-monomers, we observe some C₄-2N-monomers with relatively high intensity, such as $C_4H_6N_2O_7$ at $m/z$ 273 and $C_4H_8N_2O_8$ at $m/z$ 291. It is proposed that such C₄ dinitrates originate from the further oxidation of C₅ carbonyl compounds followed by unimolecular decomposition (Schwantes et al., 2015; Wennberg et al., 2018), as shown in Scheme S11.

2N-Dimers are C₁₀ compounds with 8-12 oxygen atoms ($C_{10}H_{16}N_2O_{8-12}$), and their signal intensities are relatively low compared to that of monomers, approximately three orders of magnitude lower. They might be ROOR products from the self or cross reaction of two nitrooxy peroxy radicals (Berndt et al., 2018). 3N-Dimers are molecules consisting of 12–16 oxygen atoms ($C_{10}H_{17}N_3O_{12-16}$). They are probably formed from further oxidation of 2N-dimers or from the cross reaction of a nitrooxy peroxy radical with a dinitrooxy peroxy radicals.



## 3.2 Multi-generation chemistry

### 3.2.1 Molecular composition for each step

As mentioned in Sect. 2.2, there were four injections during the experiment on 8 August (denoted as step I, II, III, IV in Fig. 3), wherein in the first three injections all components, $O_3$, $NO_2$, and isoprene, were added, while in the last step only $O_3$ and $NO_2$ were injected to promote the further oxidation of early-generation products. The extended oxidation time with reinjection of oxidants provides the opportunity to investigate the multi-generation oxidation chemistry of isoprene-$NO_3$ system. The mass spectra show only slow changes in the concentrations during the last period of each step, indicating weak chemical evolution. Therefore, we use integrated mass spectra over the last 10 minutes of each step for further analysis. Due to the similarity of the integrated mass spectra for step II and step III, the latter is omitted in Fig. 3.

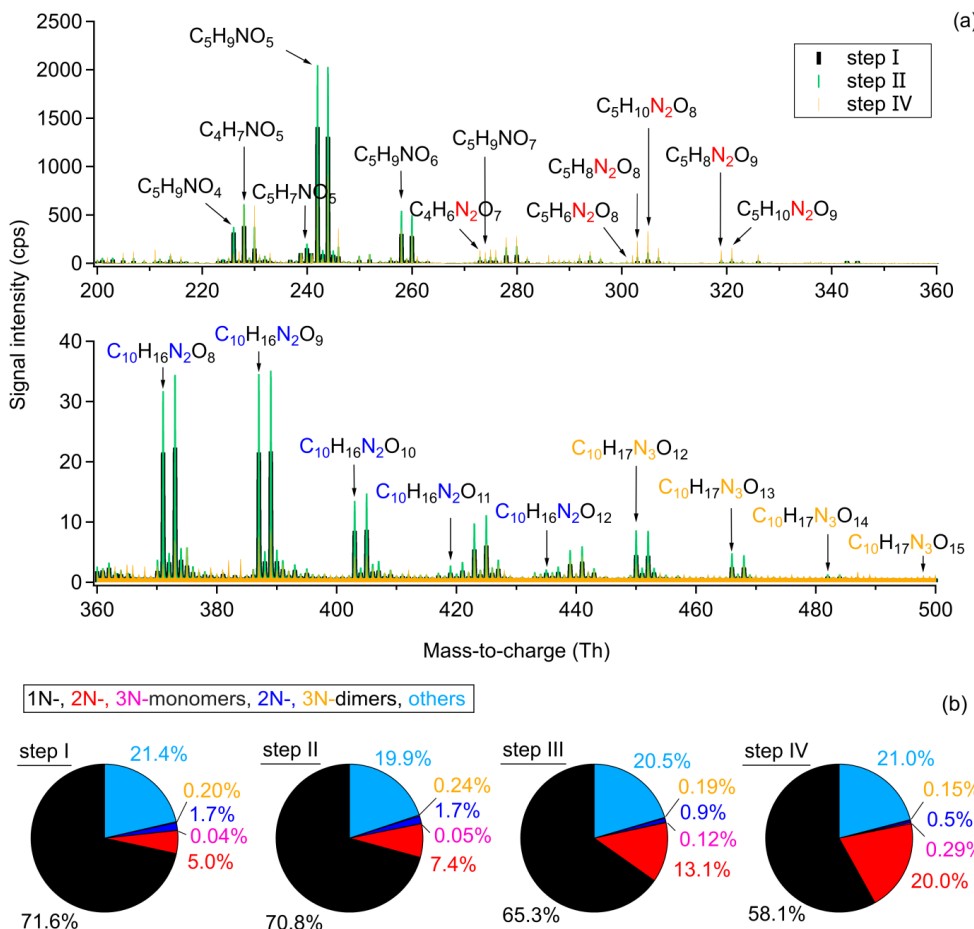

**Figure 3: Comparison of the chemical composition of each oxidation step. (A) Averaged mass spectra for step I, II, and IV, with the omitted spectrum of step III being very similar to that of step II. (B) Relative contribution of different chemical groups for each oxidation step. Only organic products were counted for analysis. 'Others' refers to CHO compounds without containing nitrogen atoms (e.g., $C_5H_8O_2$ and $C_5H_8O_3$).**



As shown in Fig. 3a, large amounts of 1N-monomers were formed from $NO_3$ oxidation of isoprene in step I,
wherein $C_5H_9NO_5$, $C_5H_9NO_6$, and $C_4H_7NO_5$ are the most abundant compounds in signal. The 2N-monomers,
which are expected from further oxidation of 1N-monomers, are much less compared to 1N-monomers,
accounting for 5.0% of the total organic signals, with the 3N-monomers even less (0.04%). The low
contributions of second-generation products probably results from the relatively high concentration of isoprene
in step I, reducing the possibility for further oxidation of first-generation products. These results indicate that the
system is dominated by first-generation chemistry at the early stage and therefore the oxidation state of products
is low. In addition to monomers, some 2N- and 3N-dimers are observed. They contribute 1.7% and 0.2%,
respectively, to the total organic signals, as shown in Fig. 3b. The low signal intensity of dimers probably results
from their small yield under our experimental conditions. In this case their contribution to SOA formation might
be small. However, a part of the dimers condense onto chamber wall due to their low volatility, so only a
smaller portion exists in the gas phase (compare Table S3 and Fig. S6).
In step II, the secondary chemistry was accelerated by further addition of $O_3$ and $NO_2$, but the primary
chemistry was also maintained by isoprene injection. As a result, more 1N-monomers (e.g. $C_5H_9NO_{4,5,6}$) were
formed compared to step I, as well as dimers (e.g., $C_{10}H_{16}N_2O_{8,9,10}$ and $C_{10}H_{17}N_3O_{12,13}$), as shown in Fig. 3a. The
signals of 2N-monomers almost double in this period compared to those in step I, and their relative contribution
increase from 5.0% to 7.4%. This is attributed to the further oxidation of first-generation products formed in
step I. The relative contributions of different chemical groups exhibited in Fig. 3b clearly show that, although
$NO_3$ produced from the second addition of $NO_2$ and $O_3$ still primarily reacted with newly-injected isoprene,
reaction of $NO_3$ with the first-generation oxidation products retaining a double bond was inevitable, leading to
more second-generation 2N- or 3N-products compared to step I. The visibly increasing fraction of 2N-
monomers indicates that the second-generation chemistry started to play a more important role than that in the
early stage. In step III, the chemical process proceeded similarly, and thus is not further discussed here.
Due to the favorable conditions for further oxidation, the signals of 1N-monomers (such as $C_5H_9NO_4$,
$C_5H_9NO_5$, and $C_5H_9NO_6$), as well as 2N- and 3N-dimers, dropped dramatically in step IV, with their relative
contributions decreasing to 58.1%, 0.5%, and 0.15%, respectively. The decrease in signals of dimers is primarily
ascribed to lack of isoprene, as there were less peroxy radicals under this condition, and hence less dimers were
formed. In addition, their condensation on the wall and dilution also contributed to the decreasing signals.
Furthermore, dimers with 2 or 3 nitrogen atoms possess at least one double bond in their molecular structures
and can thus be further oxidized under high $NO_3$ condition to form 4N- or 5N-dimers. However, only few 4N-
dimers and no 5N-dimers were detected by CIMS, suggesting that the 4N- and 5N-dimers were either not
formed, or condensed on the wall due to their low volatilities. In contrast, 2N- and 3N-monomers increase
significantly, with their relative contributions ascending to 20.0% and 0.29%, respectively. This indicates that
2N- and 3N-monomers might be second- or later-generation products that are formed from the further oxidation
of first-generation products. Additionally, unlike the $C_5$ monomers, the signal of $C_4H_7NO_5$ increased in step IV,
indicating that there is a new formation pathway for $C_4H_7NO_5$ under excess $NO_3$ condition. No double bond can
remain in such products, as otherwise they would be oxidized and their signal should decay instead.
In summary, above findings confirm that multi-generation chemistry happened during the $NO_3$-initiated
isoprene oxidation, and that the later generation oxidation was promoted by "excess" $NO_3$ radicals.





### 3.2.2 Carbon oxidation state ($\overline{OS_C}$)

The oxidation state of carbon ($\overline{OS_C}$) is defined as the charge a carbon atom takes with assumption that it loses completely all electrons in bonds to more electronegative atoms and vice versa (Kroll et al., 2011). This quantity is a metric for the degree of oxidation and will increase with oxidation. Moreover, $\overline{OS_C}$ together with carbon number can be used to constrain the composition of organic mixtures and provide insights into their evolutions. The carbon oxidation state of a species is determined by the relative abundances and oxidation states of non-carbon atoms in the compound. Since we observed nitrate groups in the products, $\overline{OS_C}$ is defined by Eq. (3). In this study, the group-averaged $\overline{OS_C}$ is the signal-weighted mean average carbon oxidation state of compounds with the same carbon number, and the bulk-averaged $\overline{OS_C}$ is the signal-weighted mean average carbon oxidation state of all detected compounds in the system.

$$\overline{OS_C} = \frac{2 \times n_O - n_H - 5 \times n_N}{n_C} \qquad (3)$$

wherein, $n_O$, $n_H$, and $n_N$ are the number of the respective atoms in the molecular formula.

Figure 4 shows the distribution of gas-phase products from the isoprene-$NO_3$ system in the oxidation state versus carbon number ($OS_C$ vs $n_C$) space. The bulk-averaged $\overline{OS_C}$ is -0.35 in step I, wherein the smaller molecules ($C_{\leqslant 4}$) have higher oxidation states than the larger molecules. The group-averaged oxidation state of $C_5$ compounds is relatively low ($\overline{OS_{C=5}}$ = -0.66), indicating that both of the oxidation and autoxidation degree of isoprene are quite low during this period. This is consistent with the conclusion made previously from mass spectra results that at the early stage isoprene-$NO_3$ oxidation was dominated by first-generation chemistry.

The system $\overline{OS_C}$ increases to -0.26 in step II, confirming that first-generation products were further oxidized after the second injection. During this step, the $\overline{OS_C}$ of most compound groups increase only weakly, except for that of the $C_5$ compounds. The group-averaged $\overline{OS_C}$ of $C_5$ compounds increases to -0.60 in step II, which is the major contributor to the increase of $\overline{OS_C}$ of the whole system. The increase of $\overline{OS_C}$ of $C_5$ compounds is largely attributed to the formation of 2N-monomers expected from further oxidation of existing 1N-products formed in step I. This is confirmed by the detectable increase of 2N- and 3N-monomers in the mass spectra and their higher relative contributions to total signals (see Fig. 3). In addition to $C_5$ compounds, the $\overline{OS_C}$ of $C_3$ and $C_6$ products increase significantly in step II.

In step IV, the secondary oxidation was largely accelerated by reinjection of $O_3$ and $NO_2$, and hence the system oxidation degree increases, with the bulk-averaged $\overline{OS_C}$ growing substantially to 0.09. Similarly, the significant increase of system $\overline{OS_C}$ is mainly attributed to the $C_5$ compounds, with their group-averaged $\overline{OS_C}$ increasing to -0.31. In addition, the $\overline{OS_C}$ of $C_{10}$ compounds increased evidently despite their decreasing signals, suggesting $C_{10}$ dimers were further oxidized as well in step IV. It is worth noting that the average carbon number decreases step by step with increasing $\overline{OS_C}$. This is the case because fewer $C_{10}$ products, but more fragments were formed with the reaction proceeding, as shown in Fig.4 by the decreasing peak areas of larger molecules but converse trend for smaller molecules. One conceivable explanation for the decreasing dimers but increasing fragments with the increasing $\overline{OS_C}$ is that, with more highly oxidized $RO_2$ formed under high $NO_3$ condition, the prevailing fate of $RO_2$ changes from dimerization to forming alkoxy radicals, which would undergo unimolecular decomposition rapidly, especially when there is a neighboring oxygen-containing functional group (Molteni et al., 2019).



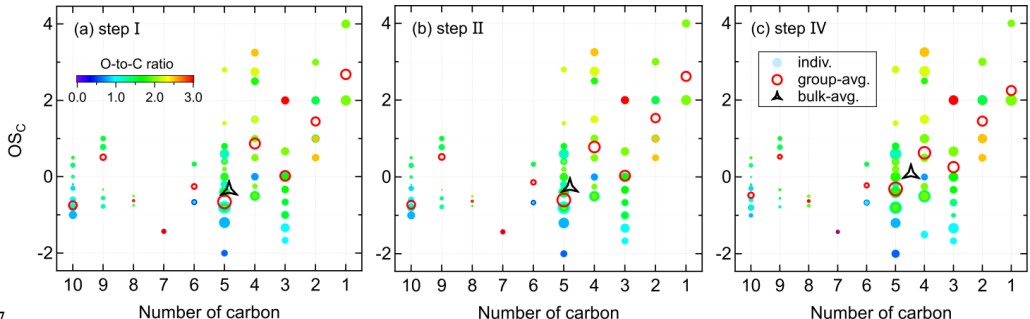

**Figure 4: Distribution of gas-phase products from isoprene oxidation by NO₃ in the carbon oxidation state ($OS_C$)**
**versus carbon number ($n_C$) space. Markers are colored by oxygen-to-carbon molar ratio and sized by the logarithm**
**of peak areas. The group-averaged and bulk-averaged $\overline{OS_C}$ are signal-weighted mean average carbon oxidation state**
**of compounds with the same carbon number and of all detected compounds, respectively.**

In summary, isoprene and its products undergo further oxidation by NO₃, leading to an increase in degree

of oxidation of products as the reaction proceeds. The increasing bulk $\overline{OS_C}$ is largely governed by the highly

oxidized C₅ compounds. In addition, more fragments but fewer dimers are formed as the $\overline{OS_C}$ increases, which

can be probably explained by the change of RO₂ fate from prevailing dimerization to fragmentation through the

alkoxy radical channel.

### 3.2.3 Characteristics of different-generation products

**(1) 1N-monomers**

To illustrate the multi-generation chemistry involved in the isoprene-NO₃ reaction system, Fig. 5 shows the time

evolution of the major gas-phase products. The signal of the most abundant compounds, C₅H₉NO₅, increases

rapidly as soon as the reaction was initiated, reaching a maximum when its chemical production rate matches its

loss rate (including chemical destruction, wall loss, dilution, etc.), and decreases slowly thereafter. Its time

behavior in the first three steps is similar. In step IV, however, the injection of O₃ and NO₂ resulted in a strong

decay of C₅H₉NO₅, owing to the occurrence of further oxidation by NO₃. The time behavior suggests that

C₅H₉NO₅ signal is dominated by first-generation oxidation products, and the same conclusion can be made for

C₅H₉NO₄ and C₅H₉NO₆. According to the mechanistic framework developed above, the C₅H₉NO₄, C₅H₉NO₅,

and C₅H₉NO₆ compounds most likely correspond to hydroxyl nitrates, nitrooxy hydroperoxides, and hydroxy

hydroperoxy nitrates, respectively, but other constitutional isomers are possible. They were already observed in

previous studies and were proposed to form through reactions of INO₂ radicals with RO₂, HO₂, and

unimolecular rearrangement, as shown in Scheme S5 (Ng et al., 2008; Kwan et al., 2012; Schwantes et al., 2015;

Wennberg et al., 2018).



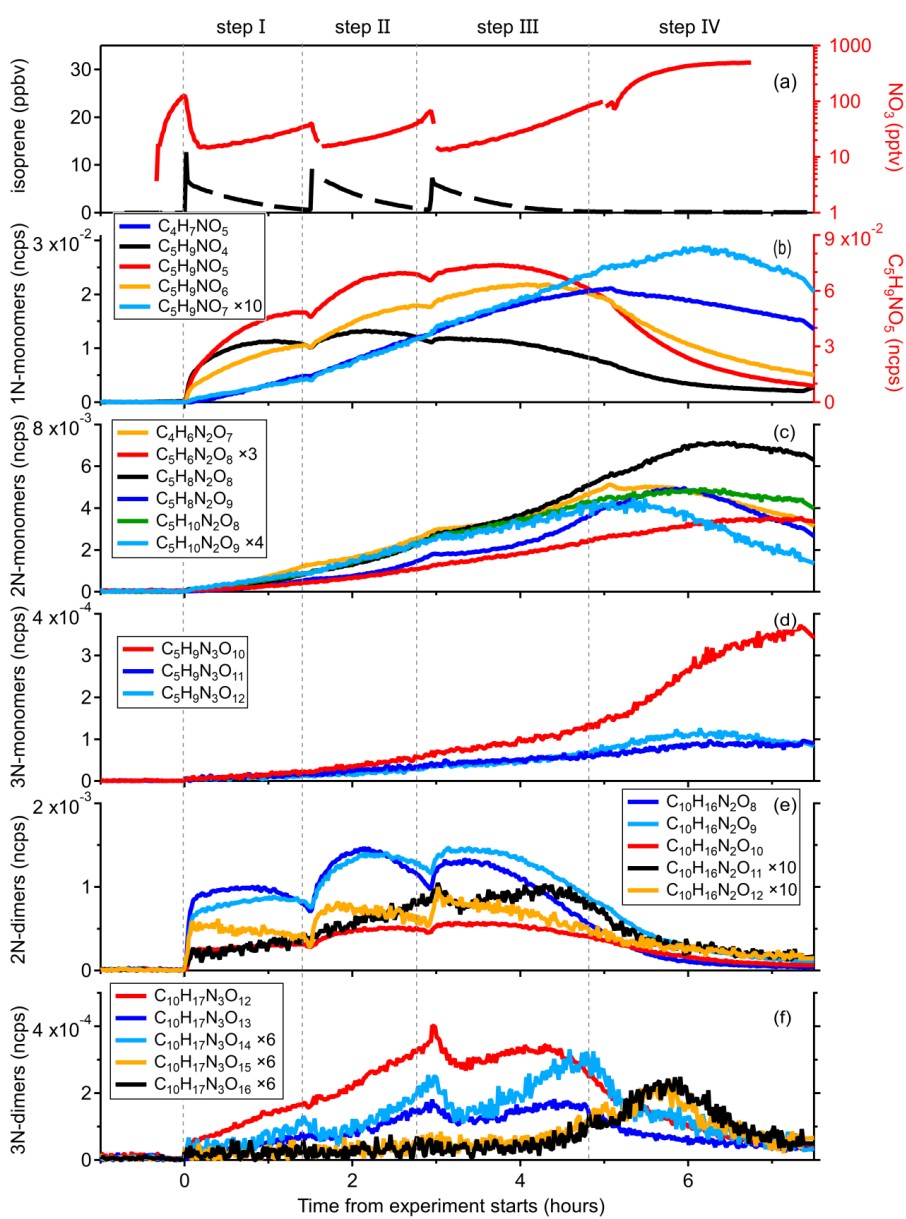

**Figure 5: Time evolution of selected gas-phase compounds measured during the isoprene - NO$_3$ experiment on 08 August, 2018. (a) Time series of O$_3$, NO$_2$, NO$_3$ and isoprene. (b)–(f) Time evolution of major 1N-monomers (C$_5$H$_9$NO$_{4-7}$ and C$_4$H$_7$NO$_5$), 2N-monomers (C$_4$H$_6$N$_2$O$_7$, C$_5$H$_6$N$_2$O$_8$, and C$_5$H$_{8,10}$N$_2$O$_{8,9}$), 3N-monomers (C$_5$H$_9$N$_3$O$_{10-12}$), 2N-dimers (C$_{10}$H$_{16}$N$_2$O$_{8-12}$), and 3N-dimers (C$_{10}$H$_{17}$N$_3$O$_{12-16}$).**

As shown in Fig. 5b, the temporal evolution of C$_5$H$_9$NO$_7$ (*m/z* 274) is different to C$_5$H$_9$NO$_{4-6}$ compounds,

suggesting that it has a completely different formation pathway. Specifically, the formation rate of C$_5$H$_9$NO$_7$ is

initially much slower than that of C$_5$H$_9$NO$_{4-6}$ but accelerates to become comparable to them later as the

experiment proceeds, i.e. when a multitude of first-generation products are accumulated. This implies that





$C_5H_9NO_7$ is produced from the further oxidation of first-generation products, and its signal is dominated by
second-generation products. Based on its molecular composition, $C_5H_9NO_7$ could be the dihydroperoxy nitrate
as shown in Scheme S5, but its formation through the reaction of $HO_2$ with nitrooxy hydroperoxy radical from
$INO_2$ autoxidation suggests it should be first-generation products, not in accordance with the time behavior we
actually observe. Consequently, we can conclude that it is not the major formation pathway that contributed to
$C_5H_9NO_7$ observed in this study. As shown in Scheme S7, the first-generation $C_5$ hydroxy carbonyl ($C_5H_8O_2$,
$m/z$ 179) can be further oxidized by $NO_3$ and the resulting alkyl radical would rapidly recombine with $O_2$,
producing a new peroxy radical, which then reacts with $HO_2$ radicals to form $C_5H_9NO_7$ (hydroxy hydroperoxy
carbonyl nitrate). Similarly, the $C_5$ hydroperoxy carbonyl ($C_5H_8O_3$, $m/z$ 195) can also lead to the formation of
such $C_5H_9NO_7$ (isomer of that formed through $C_5H_8O_2$ channel) through further oxidation (see Scheme S7).
According to above two mechanisms, $C_5H_9NO_7$ formed following such reaction pathways should be second-
generation products, better consistent with its time behavior.

Considering its similar time behavior to $C_5H_9NO_7$, the observed $C_4H_7NO_5$ ($m/z$ 228) signal is likewise
thought to be dominated by second-generation products. Schwantes et al. (2015) proposed such a $C_4$ product
based on OH-initiated chemistry, but as the OH concentration in our system was close to zero during the
experiment (see Fig. S3), this formation pathway cannot apply in our situation. Instead, we suggest that
$C_4H_7NO_5$ is formed through the unimolecular decomposition of the $C_5$ alkoxy or acyl radicals, which result from
further oxidation of the $C_5$ hydroxy carbonyl ($C_5H_8O_2$, $m/z$ 179), as shown in Scheme S7. It should be pointed
out here that there may be reaction pathways forming $C_4H_7NO_5$ as first-generation products that are not
considered here, whereas it is no doubt that the second-generation chemistry played a dominant role in $C_4H_7NO_5$
formation according to its time evolution measured by CIMS.

Although $C_4H_7NO_5$ and $C_5H_9NO_7$ show similar time behaviors in the first three steps, it seems that they
followed fairly different reaction pathways when the concentration of $NO_3$ in the chamber increased
dramatically in step IV. As shown in Fig. 5b, the signal of $C_4H_7NO_5$ drops immediately after the injection of $O_3$
and $NO_2$, while that of $C_5H_9NO_7$ continues to increase, although its formation rate becomes slightly lower with
increasing $NO_3$ concentration. The decay of $C_4H_7NO_5$ signal can be explained by more chemical destruction or
less production under high $NO_3$ condition, wherein the latter seems more sensible in terms of its structure (no
double bond remaining). As shown in Scheme S7, the second-generation $C_4H_7NO_5$ and $C_5H_9NO_7$ compounds
share the same precursor in the $C_5H_8O_2$ channel. Consequently, the production of $C_5H_9NO_7$ through this
pathway would be interrupted immediately after the injection of $O_3$ and $NO_2$ like $C_4H_7NO_5$. In reality, its signal
might decay even faster due to the larger reaction rate of $RO_2$ H-shift (leading to the formation of $C_4H_7NO_5$)
than that of $RO_2$ reacting with $HO_2$ (leading to the formation of $C_5H_9NO_7$). As presented by Vereecken and
Nozière (2020), the rate coefficient of aldehydic H-shift is $\geq 0.5$ s$^{-1}$ (298 K), while the pseudo first order rate
coefficient of $RO_2$ reacting with $HO_2$ is ~ $10^{-3}$ s$^{-1}$ ($k$ (298 K) = 5 ×$10^{-12}$ cm$^3$ molecules$^{-1}$s$^{-1}$ (Atkinson, 2007), and
[$HO_2$] ~ 4 ×$10^8$ molecules cm$^{-3}$), about two orders of magnitude smaller. This result implies that the increasing
$C_5H_9NO_7$ observed is contributed to by other formation pathways. As mentioned before, $C_5H_9NO_7$ can also be
produced by $C_5H_8O_3$ oxidation. We find that the signal of $C_4H_7NO_6$ ($m/z$ 244), which results from $C_5H_8O_3$
oxidation as well, remains increasing after the injection of $O_3$ and $NO_2$. This tentatively confirms that the
production of $C_5H_9NO_7$ in step IV is mainly from $C_5H_8O_3$ oxidation channel. More experimental or theoretical
studies are needed to provide insights into these differences.





**(2) 2N- and 3N-monomers**

As shown in Fig. 5c, 2N-monomers formed much slower than 1N-monomers in the early stage, but their formation rates were accelerated in step II and step III, probably due to the accumulation of first-generation products. According to our mechanistic framework, 2N-monomers are second-generation products resulting from the further oxidation of 1N-monomers by $NO_3$, which is consistent with their time behaviors detected by CIMS.

Like $C_4H_7NO_5$ and $C_5H_9NO_7$, different 2N-monomers have similar behavior in the first three steps, but they are obviously different in step IV when the concentration of $NO_3$ increased drastically in the chamber. For instance, the signals of $C_5H_8N_2O_8$, $C_5H_8N_2O_9$ and $C_5H_{10}N_2O_8$ continue to increase after the injection of $O_3$ and $NO_2$, while that of $C_5H_{10}N_2O_9$ drops immediately. This is related to their detailed formation mechanisms which are outside the scope of this study. Furthermore, $C_5H_8N_2O_9$ and $C_5H_{10}N_2O_9$ decay a little bit faster than $C_5H_8N_2O_8$ and $C_5H_{10}N_2O_8$, which might be related to their volatility and will be further discussed in next section.

Different from other 2N-monomers, the signals of $C_5H_6N_2O_8$ (*m/z* 301) increases continuously under high $NO_3$ condition, although its net formation rate is almost zero at the end of step IV. The characteristics of $C_5H_6N_2O_8$ under high $NO_3$ condition reflects its different formation pathways from other dinitrates, and without having a comprehensive knowledge of its chemical mechanism, we are unable to tell what exactly leads to the differences. In the Master Chemical Mechanism (MCM v3.3.1), $C_5H_6N_2O_8$ is proposed to be a PAN-like compound stemming from the $C_5$ nitrooxy carbonyl ($C_5H_7NO_4$) (http://mcm.leeds.ac.uk/MCM/browse.htt?species=NC4CHO). Such $C_5H_6N_2O_8$ compound would react with $NO_3$ radicals due to the remaining double bond, and hence this cannot be the predominant formation pathway of the $C_5H_6N_2O_8$ observed in this study. Based on the formation mechanism of dinitrooxyepoxides ($C_5H_8N_2O_7$) proposed by Kwan et al. (2012), we suggest that $C_5H_6N_2O_8$ can also be a dinitrooxyepoxide resulting from cyclization of specific hydroperoxy alkyl radicals, as shown in Scheme S10. Alternatively, the $C_5$ hydroxy nitrate ($C_5H_9NO_4$) can be oxidized by $NO_3$ and then react with $NO_3$ radicals again, forming $C_5H_6N_2O_8$ with two aldehyde groups ultimately (see Scheme S10). According to the proposed mechanisms above, $C_5H_6N_2O_8$ formed through the first two pathways are second-generation products, while those from the third channel are third-generation products, in accordance with its time behavior measured by CIMS.

In addition to $C_5$-2N-monomers, we observe some $C_4$ dinitrates such as $C_4H_6N_2O_7$ (*m/z* 273) and $C_4H_8N_2O_8$ (*m/z* 291), and the signal intensity of $C_4H_6N_2O_7$ is comparable to the major $C_5$-2N-monomers. $C_4$ dinitrates have rarely been mentioned in previous isoprene-$NO_3$ studies. As shown in Fig. 5c, $C_4H_6N_2O_7$ has similar time behavior to $C_5$-2N-monomers, and hence is thought to be second-generation products. Wennberg et al. (2018) proposed that such a $C_4$ dinitrate was generated from OH-initiated further oxidation of $C_5H_7NO_4$. However, this is not applicable here due to a lack of OH radicals in our system. Instead, we propose that the $C_4H_6N_2O_7$ observed in this study is dinitrooxy carbonyl compound resulting from $NO_3$ oxidation of $C_5H_7NO_4$ with subsequent unimolecular decomposition (see Scheme S11 for details).

As shown in Fig. 5d, 3N-monomers are generated more slowly than 1N-monomers, but their signals grow gradually as the experiment proceeds, with a significant increase especially for $C_5H_9N_3O_{10}$ in the last step. Furthermore, we can see from Fig. 5c and Fig. 5d that the signals of $C_5$ trinitrates in step IV appear anticorrelated to that of $C_5H_{10}N_2O_8$ and $C_5H_{10}N_2O_8$. The gas-phase 3N-monomers have rarely been reported in previous literature. Ng et al. (2008) observed $C_5H_9N_3O_{10}$ compound in the particle-phase and assumed that it



was produced from $NO_3$ oxidation of the $C_5$ hydroxy nitrate ($C_5H_9NO_4$). Similarly, $C_5H_9N_3O_{11}$ and $C_5H_9N_3O_{12}$
can be formed through $NO_3$ reacting with dinitrooxy peroxy radicals, which result from corresponding first-
generation nitrooxy compounds ($C_5$ hydroperoxy nitrate, $C_5H_9NO_5$ or $C_5$ hydroxy hydroperoxy nitrate, $C_5H_9NO_6$)
oxidation by $NO_3$ radicals, as shown in Scheme S12. 3N-Monomers formed following such pathways are
second-generation products by definition. Regarding the rising signals of 3N-monomers in step IV, one
explanation is that although the reaction of dinitrooxy peroxy radicals with $NO_3$ is not an oxidation process,
their formation can be significantly facilitated by increasing $NO_3$ concentration. It is also possible that 3N-
monomers are formed through H-abstraction of 2N-monomers. $NO_3$ radicals can abstract the hydrogen of
dihydroxy dinitrate ($C_5H_{10}N_2O_8$) or hydroxyl hydroperoxy dinitrate ($C_5H_{10}N_2O_9$) from the carbon with an –OH,
–OOH or –$ONO_2$ group attached, leading to alkyl radicals that can subsequently recombine with $O_2$ and then
react with $NO_2$ or $NO_3$, yielding trinitrates or peroxynitrates containing three nitrogen atoms. 3N-Monomers
stemming from such reactions ought to be third-generation products. However, we should point out that 3N-
monomers formed following H-abstraction pathway are less likely because abstracting hydrogen from the
hydroxyl, hydroperoxy or nitrooxy carbon would lead to fragmentation at most cases (Bianchi et al., 2019).

In addition, it is interesting to note that the signal of $C_5H_9N_3O_{10}$ increases continuously throughout step IV,

whereas that of $C_5H_9N_3O_{11}$ and $C_5H_9N_3O_{12}$ drop after a short period of growth. Meanwhile, the production of
$C_5H_9N_3O_{10}$ is facilitated by the increasing $NO_3$ concentration compared to that of $C_5H_9N_3O_{12}$ and $C_5H_9N_3O_{11}$.
Currently, we cannot explain what exactly causes these differences, but we suspect that there may be different
chemical pathways forming different 3N-monomers that are not covered here and may also be related to their
different physical properties, such as vapor pressures.
**(3) 2N- and 3N-dimers**
As shown in Fig. 5e, 2N-dimers (except for $C_{10}H_{16}N_2O_{11}$) display very similar time behavior to 1N-monomer,
which form rapidly after each injection, indicating that the signals of 2N-dimers are dominated by first-
generation products like most 1N-monomers. It is noted that the time behavior of $C_{10}H_{16}N_2O_{11}$ (*m/z* 419) is
completely different from that of other 2N-dimers. As illustrated in Fig. 5e, the production rate of $C_{10}H_{16}N_2O_{11}$
is initially much slower compared to other dimers. Besides, its signal increases monotonically in the first two
oxidation stages, whereas that of the others always increase first, approaching the maximum as its chemical
production competes against the losses, and decrease gradually thereafter. The special time behavior of
$C_{10}H_{16}N_2O_{11}$ suggests that it has a different formation pathway from other 2N-dimers, and its signal is most
likely dominated by secondary products. In addition, we find that the signal of $C_{10}H_{16}N_2O_{12}$ always starts to
decay earlier than that of $C_{10}H_{16}N_2O_8$ and $C_{10}H_{16}N_2O_9$. If we assume that their production rates have the same
order of magnitude (confirming by their formation rates after each injection), then it can be concluded that
$C_{10}H_{16}N_2O_{12}$ had additional chemical destruction, or its volatility is much lower than $C_{10}H_{16}N_2O_8$ and
$C_{10}H_{16}N_2O_9$ and hence has more rapid lost on the wall. It seems the second hypothesis is more likely when
comparing its signal with and without dilution and wall-loss corrections (see Fig. S4). More detailed discussion
about volatilities of different isoprene organonitrates will be provided in the next section.

It is proposed that dimers (ROOR') are likely formed through the self- or cross-reaction of two peroxy

radicals (Berndt et al. 2018). Consequently, the generation number of dimers depends only on how the involved
peroxy radicals are formed. Table S1 summarizes the possible permutation scheme of 2N-dimers from $RO_2$ +


$RO_2^{\cdot}$ reactions, and their structural information can be found in Scheme S13. For example, self-reaction of two
$C_5$ nitrooxy peroxy radicals ($C_5H_8NO_5$) leads to the formation of $C_{10}H_{16}N_2O_8$ compound, while recombination
of two $C_5$ nitrooxy hydroxyl peroxy radicals ($C_5H_8NO_6$) or a $C_5$ nitrooxy peroxy radical ($C_5H_8NO_5$) with a $C_5$
nitrooxy hydroperoxy peroxy radical ($C_5H_8NO_7$) results in $C_{10}H_{16}N_2O_{10}$ compound. According to their time
behavior, 2N-dimers (except for $C_{10}H_{16}N_2O_{11}$) are thought to be first-generation products, and from this fact we
can infer that the peroxy radicals contributing to dimer formation are dominated by first-generation
intermediates. With regard to $C_{10}H_{16}N_2O_{11}$, we conclude that it is most likely a secondary product considering
its typical second-generation behavior. In other words, at least one of the two $C_5$ nitrooxy peroxy radicals
involved in formation of $C_{10}H_{16}N_2O_{11}$ must be a secondary intermediate. As listed in Table S1, $C_{10}H_{16}N_2O_{11}$ can
be formed through $C_5H_8NO_6$ + $C_5H_8NO_7$ or $C_5H_8NO_6$ + $C_5H_8NO_7$ reactions, wherein $C_5H_8NO_7$ and $C_5H_8NO_8$
would be secondary peroxy radicals if they are formed through $NO_3$ further oxidation of the $C_5$ hydroxy
carbonyl compounds ($C_5H_8O_2$ or $C_5H_8O_3$), as shown in Scheme S7. In addition, it is possible that $C_{10}H_{16}N_2O_{11}$ is
formed from a $C_5$ hydroxy peroxy radical $C_5H_9O_3$ reacting with a $C_5$ dinitrooxy hydroxy carbonyl peroxy radical
$C_5H_7N_2O_{10}$ (from $C_5H_7NO_5$ oxidation by $NO_3$), as we observe high abundant $C_5H_{10}O_3$ during the experiment,
although $C_5H_{10}O_3$ is assumed to be the major product of the OH-initiated chemistry.
Apart from 2N-dimers, we observe detectable signals at *m/z* 450, 466, 482, 498 and 514, which are
identified as 3N-dimers with molecular formulas $C_{10}H_{17}N_3O_{12-16}$. $C_{10}H_{17}N_3O_{12}$ and $C_{10}H_{17}N_3O_{13}$ were detected
in the particle-phase in previous study, suggesting that they have low volatility and can contribute to SOA
formation (Ng et al., 2008). As shown in Fig. 5f, 3N-dimers form much slower than 2N-dimers, but their
productions are accelerated as the experiment proceeds. This is similar to the characteristics of second-
generation 2N- and 3N-monomers to some degree, suggesting that the signals of 3N-dimers we observed are
most likely dominated by secondary or even later-generation compounds.
It is worth noting that $C_{10}H_{17}N_3O_{12-14}$ and $C_{10}H_{17}N_3O_{15,16}$ have two completely different types of time
behavior. The signals of $C_{10}H_{17}N_3O_{12}$, $C_{10}H_{17}N_3O_{13}$ and $C_{10}H_{17}N_3O_{14}$ more or less increase in the first three
oxidation steps and start to decline in the late of step III with increasing $NO_3$ concentration. As depicted in
Scheme S13, 3N-dimers can result from further oxidation of 2N-dimers or the cross-reaction of a first-
generation nitrooxy peroxy radical with a secondary dinitrooxy peroxy radical. Accordingly, such 3N-dimers are
thought to be second-generation products, and they would further react with $NO_3$ due to the remaining double
bond in their molecular structure, leading to severe chemical destruction of these compounds under high $NO_3$
condition. This is consistent with the time behavior of $C_{10}H_{17}N_3O_{12}$, $C_{10}H_{17}N_3O_{13}$ and $C_{10}H_{17}N_3O_{14}$. In contrast,
$C_{10}H_{17}N_3O_{15}$ and $C_{10}H_{17}N_3O_{16}$ are formed even more slowly, and their production in the first four hours is close
to zero. However, their signals start to climb in the late of step III, during which that of $C_{10}H_{17}N_3O_{12}$,
$C_{10}H_{17}N_3O_{13}$ and $C_{10}H_{17}N_3O_{14}$ decline. This suggests that $C_{10}H_{17}N_3O_{15}$ and $C_{10}H_{17}N_3O_{16}$ formed under high $NO_3$
condition probably result from further reactions of $C_{10}H_{17}N_3O_{12-14}$. However, this assumption is highly uncertain
and more experimental and theoretical studies are needed to substantiate it. In terms of their time behavior,
$C_{10}H_{17}N_3O_{15}$ and $C_{10}H_{17}N_3O_{16}$ are thought to be third- or even later-generation products.


**3.3 Volatility distribution of isoprene nitrates**
**3.3.1 $C^*$ estimated by experimental methods**
Detailed information about the volatility of organic molecules is essential to evaluate their potential to form
SOA. In order to investigate the potential contribution of various isoprene oxidation products to SOA formation,
we use our (limited) experimental data to estimate the vapor pressure of different isoprene organonitrates on the
basis of their condensation behavior. Figure 6 shows how the signals of gas-phase products change in
experiments with and without seed aerosols (ammonium sulfate). Please note that while the two experiments
were conducted under similar conditions, the procedures could not be kept fully identical as aerosol seeding
required specific measures and the oxidation chemistry might be slightly altered (e.g., due to initiation of
heterogeneous reactions).
As shown in Fig. 6, the signals of most of the selected compounds decline when there are seed aerosols in
the chamber, indicating that part of the condensable vapors is partitioned to the particle-phase due to the
introduction of condensation sinks. The decrease in signal differs for different products, mostly depending on
their vapor pressures. As expected, the lower volatility of a compound the higher the fraction that condenses.
For instance, the signal of $C_5H_9NO_7$ decreases by more than 70% in experiment with seed aerosols, compared to
less than 40% on average for other less-oxidized 1N-monomers. In some cases (e.g., $C_5H_9NO_4$ and $C_5H_9NO_5$)
however, the product signals in experiment with seed aerosols are higher than that without seeds after the
consumed isoprene exceeding a certain level. In addition, the signal of $C_5H_6N_2O_8$ in the experiment with seeds
is always higher compared to that without seeds. One explanation for this phenomenon is the effect of
heterogeneous reactions. It is likely that some condensed compound (denoted as A) can react on the particle
surface to form new products with the molecular composition of compound B, or alternatively forming a
precursor of B. When they evaporate back to the gas phase, it can result in an increase in signal of compound B.
That's why a higher signal was observed for such compounds in experiment with seeds than that without seeds,
as observed for $C_5H_6N_2O_8$ in this case.
Based on the observed condensation behavior of different products, we can derive their vapor pressures
from the gas-particle equilibrium partitioning coefficients by Eq. (2). As depicted in Fig. 7, the saturation
concentrations of different organonitrates show a decreasing tendency from 1N-, 2N-monomer and 3N-
monomers to 2N- and 3N-dimers, suggesting that dimers have a higher propensity of condensation and
contribute to SOA formation. This is partly related to their molecular weight, as larger molecules generally have
lower vapor pressures. However, it cannot explain all the features of the volatility distribution. For example,
$C_5H_9NO_6$ (corresponding to No.8 in Fig.7) has higher mass than $C_5H_9NO_5$ (corresponding to No.7 in Fig.7) but
is predicted to have higher vapor pressure. In general, chemical composition and functionalities have significant
effects on vapor pressure. For instance, the 2D-VBS composition-activity relationship suggests that each carbon
and oxygen decrease $C^*$ by 0.475 and 1.75 decades, respectively (Donahue et al., 2011). Different functional
groups also have very different effect on volatility. For example, each hydroxyl group (–OH) or hydroperoxy
group (–OOH) typically reduces the volatility by 2.4 to 2.5 decades, while the less polar carbonyl group (=O)
reduces the volatility by 1 decade (Pankow and Asher 2008, Donahue et al., 2011). The nitrooxy group (–ONO₂)
has a similar reductive effect on vapor pressure, which typically reduces $C^*$ by 2.5 orders of magnitude (Pankow
and Asher, 2008). Here, the irregularly high vapor pressure of $C_5H_9NO_6$ is most likely attributed to the
functional groups it contains. As listed in Table S2, $C_5H_9NO_6$ is proposed to be nitrooxy hydroxy hydroperoxyl

compound, which consists of two highly polar functional groups –OH and –OOH, contributing to formation of intramolecular H-bonding that can significantly increase the vapor pressure (Bilde et al., 2015; Kurten et al., 2016), while $C_5H_9NO_5$ only contains a –OOH group and hence cannot form intramolecular H-bonding. These findings underline that the constitutional and configurational information of a molecule is critical for vapor-pressure estimation.

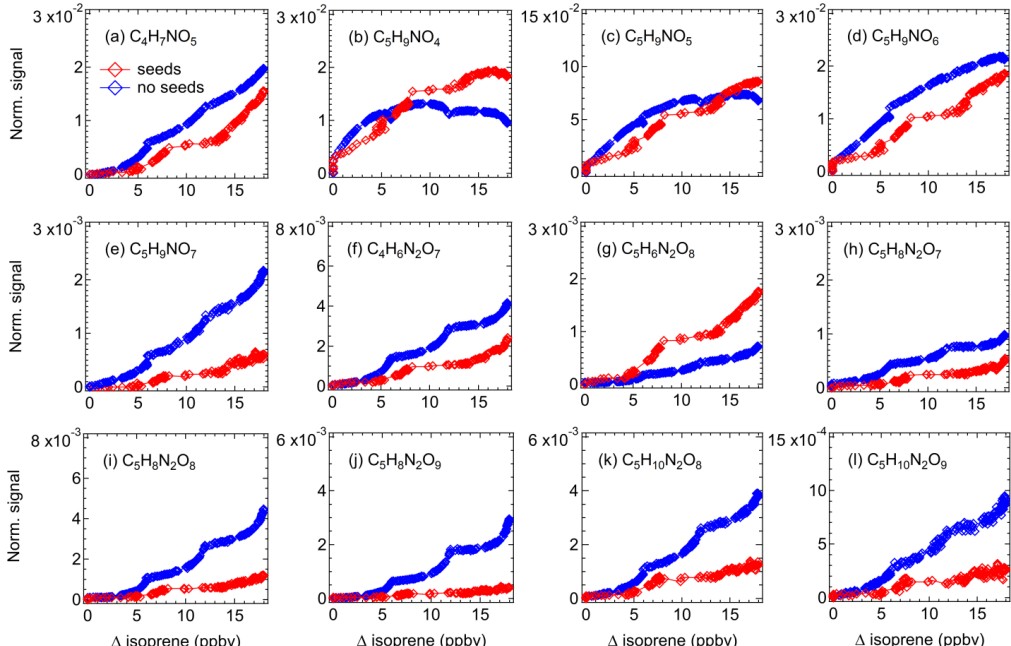

**Figure 6: Time evolution of selected major gas-phase products during experiments with (red) and without (blue) seed aerosols (ammonium sulfate). Signals have been corrected for dilution.**

### 3.3.2 $C^*$ estimated by different parametrization methods

For comparison, we also adopt different parameterization methods to estimate the saturation vapor pressures of isoprene oxidation products based on their molecular composition and the proposed structures, with the results depicted in Fig. 7. In general, the saturation concentrations calculated by different parameterization methods show a similar volatility distribution to that calculated by experimental method, with $C^*$ of 1N-, 2N- and 3N-monomers, 2N- and 3N-dimers decreasing in turn. However, different parameterization methods lead to the predicted vapor concentrations with a variability of several orders of magnitude for the same compound, and the discrepancies become larger and larger with more complicated molecules. In addition, $C^*$ of structural isomers calculated by the same method could span several decades.

As shown in Fig. 7, the Donahue et al. parameterization mostly provides lower $C^*$ compared to the three GC methods, with a maximum discrepancy up to 12 orders of magnitude for dimers. With regard to smaller and less oxidized 1N-monomers, predicted $C^*$ values from different methods are in relatively good agreement with each other, whereas the disagreement increases to 11 orders of magnitude for 2N- and 3N-monomers. This is mainly the case because the organic molecules were regarded as a mixture of =O and –OH functional groups in the Donahue et al. parameterization, and their relative abundance was assumed to be 1:1 (Donahue et al., 2011).





In consequence, the –OOH functional group in peroxides is treated as two –OH groups when adapting the
method proposed by Donahue et al. (2011). However, it is demonstrated that the extra oxygen in peroxy
moieties has little contribution to reduce vapor pressure (Pankow and Asher et al., 2008), hence treating –OOH
equivalent to two –OH functional groups would underestimate the vapor pressures of hydroperoxyl compounds.
Furthermore, organic compounds consisting of multiple polar functional groups (such as hydroperoxy, peroxy
acid, and peroxide functional groups) tend to form intramolecular H-bonding, which would increase the vapor
pressure (Bilde et al., 2015; Kurten et al., 2016). All these issues contribute to an underestimation of the vapor
pressures of multifunctional products when using the Donahue et al. parameterization. Mohr et al. (2019)
improved the parameterization for vapor-pressure estimation by taking the presence of –OOH functional groups
in HOM explicitly into consideration and revising the parameters to reduce the effect of –OOH on depressing $C^*$.
Consequently, the Mohr et al. parameterization effectively reduces the discrepancy between its estimates and
those predicted by the GC methods, with the differences within 6 orders of magnitude. Nevertheless, there is a
slight tendency to underestimate the vapor pressures of 3N-monomers and dimers. The Peräkylä et al.
parameterization method, which was derived from measurements of the condensation behavior of HOM
produced from α-pinene ozonolysis, predicts similar $C^*$ to Donahue et al. method for 1N-monomers, but higher
$C^*$ for 2N- and 3N-monomers like the Mohr et al. method. As for dimers, especially for the 3N-dimers
containing more multifunctional groups, the Peräkylä et al. method even predicts higher $C^*$ than the GC
methods in most cases.

Three GC methods predict similar saturation vapor pressures for different isoprene nitrates in this work,
with the differences within 5 orders of magnitudes. Generally, the SIMPOL.1 method always provides higher $C^*$
compared to another two methods, and the disagreement between methods becomes larger for molecules
containing multifunctional groups. For instance, the vapor-pressure discrepancy between SIMPOL.1 and
another two GC methods are both 2 orders of magnitude for $C_5H_9NO_{4,5}$ and $C_{10}H_{17}N_3O_{12-14}$, but it increased up
to 4 and 5 orders of magnitude, respectively, for $C_5H_9NO_{6,7}$ and $C_{10}H_{17}N_3O_{15,16}$.

It is worth noting that the Nannoolal et al. method is able to distinguish between positional isomers (e.g.,
the estimated $C^*$ for two $C_5H_{10}N_2O_9$ isomers are 0.858 and 0.333 μg m$^{-3}$, respectively), whereas such capacity of
EVAPORATION method is limited (e.g., it is able to distinguish between the position isomers of $C_5H_{10}N_2O_9$,
but it predicts identical $C^*$ for $C_{10}H_{16}N_2O_{11}$ isomers). In this respect, the SIMPOL.1 method cannot distinguish
between positional isomers at all. Moreover, SIMPOL.1 method predicts smaller differences between functional
group isomers for 1N-monomers and 3N-dimers compared to the Nannoolal et al. method and the
EVAPORATION, but there is no such regular pattern for 2N-monomers and 2N-dimers.

By comparing the results calculated by experimental method with those by different parameterization
methods, we can see that the GC methods predict lower saturation concentrations for 1N-monomers than the
experimental method, while the Donahue et al. and Peräkylä et al. method provide similar $C^*$ values. With
regard to 2N-monomers, the GC methods predict higher vapor pressures compared to the experimental method,
and the discrepancy decreases with decreasing saturation concentration. The disagreement of $C^*$ for 2N-
monomers estimated by experimental method and the Mohr et al. or Peräkylä et al. method are within 2 orders
of magnitude. In terms of low-volatility dimers, however, the vapor pressures calculated by the experimental
method were 1–3 orders of magnitude larger than that predicted by the parameterization methods except for the
Peräkylä et al. method. The Peräkylä et al. method provides the most similar predictions to the experimental
method for isoprene oxidation products in the full volatility range, with the disagreement within 1 order of
magnitude.

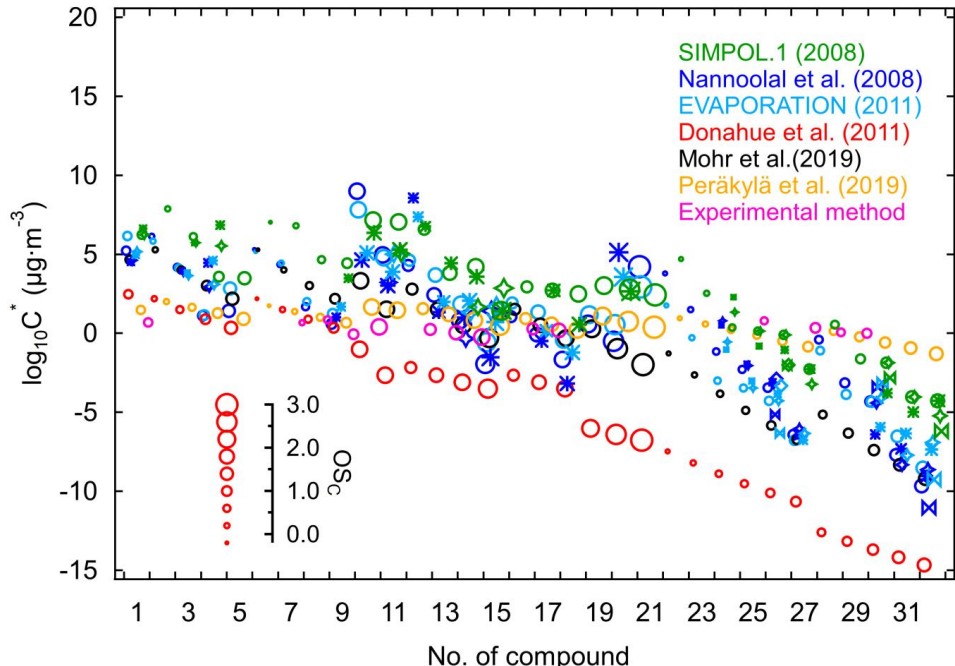


**Figure 7: Saturation concentrations (in μg m⁻³, at 298.15 K) of isoprene organonitrates estimated by using experimental and parameterization methods. The numbers correspond with the compound numbers of given in Table S2 (No. 1–9, 10–18, 19–21, 22–27, and 28–32 corresponding to 1N-monomers, 2N-monomers, 3N-monomer, 2N-dimers and 3N-dimers, respectively). Marker shapes indicate different isomers, with their size scaled by carbon oxidation state ($OS_C$).**

In general, the vapor pressures estimated experimentally in this study are very close to that calculated by
Peräkylä et al. method for which the estimation parameters were also derived experimentally. The discrepancy
between the experimental and the GC methods spans several orders of magnitude depending on different
compounds, with the GC methods predicting lower $C^*$ for less-functionalized 1N-monomers, approximate $C^*$ for
2N-monomers, but higher $C^*$ for highly functionalized dimers. It is difficult to tell which method is more
reliable without any measured saturation vapor pressure data on such multifunctional organic nitrates. However,
considering the fact that the existing GC methods tend to underestimate saturation vapor pressures of the highly
functionalized organic molecules due to their limited capability to deal with intramolecular interactions (e.g. the
intramolecular hydrogen bonding formed among polar functional groups), and the well consistent results of two
experimentally derived methods, we suggest that the experimental method might be a good choice to determine
the volatility of highly oxidized compounds accurately.
**3.3.3 Volatility distribution of isoprene nitrates and expected SOA yields.**
Although the vapor pressures calculated by different methods show a variability of several orders of magnitude,
the predicted volatility distributions of different organic groups are consistent. To eliminate the discrepancy
caused by methods and get an average trend of the volatility distribution of various isoprene nitrates, we use the





median value of C* calculated by different methods as the estimator of the vapor pressure for each nitrate
compound.

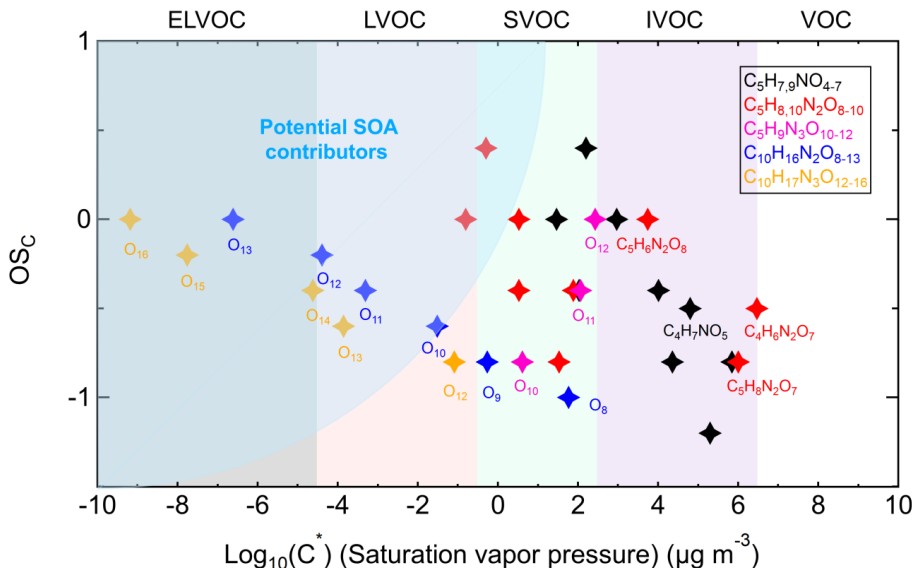

**Figure 8: Volatility distribution of different organonitrates formed from NO₃-initiated isoprene oxidation. The volatility classes are indicated along the top with corresponding colors in the plot. The position of potential SOA contributors is determined depending on the exact functionalities of molecules adapted from Bianchi et al. (2019).**

The average carbon oxidation state is plotted against $Log_{10}(C^*)$ in Fig. 8 to describe the volatility
distribution of organic nitrates formed from isoprene oxidation by NO₃. Generally, the volatility of measured
gas-phase products spans a wide range from IVOC to ELVOC, wherein all of the 1N-monomers fall in the
IVOC or SVOC range, suggesting that 1N-monomers have low potential to form SOA by simple condensation
as long as the organic aerosol load is less than 200 μg m⁻³. The addition of a second or third –NO₃ functional
group decreases C* of most 2N- and 3N-monomers by 2-3 decades compared with 1N-monomers, and most of
them belong to SVOC. They will start to condense in significant fractions if the organic aerosol load is in a
range of 1-10 μg m⁻³, which means 2N- and 3N-monomers with $OS_c$ > -0.8 may contribute to SOA formation
under atmospheric conditions. With regard to dimers, all 3N-dimers and 2N-dimers (except for $C_{10}H_{16}N_2O_{8,9}$)
are in LVOC or even ELVOC range, indicating isoprene dimers had high propensity to form SOA even at
organic aerosol loads << 1 μg/m³. However, we would like to emphasize here that the signals of 2N- and 3N-
dimers only account for less than 2% on average of the total assigned signals, as shown in Fig. S5. This suggests
that the SOA yield of isoprene from NO₃ oxidation by condensation should be low under atmospheric
conditions.
Assuming that the dimers in the LVOC or ELVOC range will condense onto particles, we estimated a SOA
mass yield for condensation of isoprene organic nitrates of about 5 %. This value is based on an averaged bulk
organonitrate sensitivity of 0.019 norm. count s⁻¹ ppbv⁻¹ and has been corrected for wall loss and dilution (see
Fig. S6, with uncorrected SOA mass yield of about 2 %). The estimated SOA mass yield is within the range of
those reported in the literature, but at the lower end (4.3% to 23.8% depending on RO₂ fate, Ng et al., 2008; 0.7%
for first generation oxidation and 14% after oxidation of both double bonds, Rollins et al., 2009; 27% on





average for ambient measurements, Fry et al., 2018). The SOA yield will probably become somewhat higher if
taking the contribution of the minor dimer products as well as SVOCs into consideration. Our finding is
commensurable with the SOA yield for isoprene organic nitrates of 2-6% derived from HR-AMS measurements
in the same campaign (Brownwood et al., in preparation).

In addition, $Br^-$ adduct ionization CIMS is selective for $HO_2$ and less oxidized organic compounds

(Albrecht et al., 2019; Rissanen et al., 2019), so it is reasonable to assume that there were more highly oxidized
products that were not detected by $Br^-$ CIMS. This assumption is confirmed by measurements with a $NO_3^-$ CIMS
performed in another isoprene-$NO_3$ experiment in SAPHIR (Zhao et al., in preparation). Zhao et al. observed a
higher fraction of dimers and more highly oxidized monomers and dimers, as well as trimers ($C_{15}$ compounds).
As a consequence, the SOA yields derived from $NO_3^-$ CIMS measurements is slightly higher.

From these points of view our yield is more a lower limit. However, even if we assume an error of a factor

of 2, the SOA yield of isoprene organic nitrates by condensation is more likely in a range of about 10% or less
than in the higher range of 20-30% published in the literature. Of course, by our method we cannot cover any
liquid phase processes that would lead to additional SOA beyond the condensation of the target compounds.
**4. Conclusions and implication**
In this work, a gas-phase experiment conducted in the SAPHIR chamber under near atmospheric conditions in
the dark was analyzed to primarily investigate the multi-generation chemistry of isoprene-$NO_3$ system. The
characteristics of a diversity of isoprene nitrates were measured by the CIMS using $Br^-$ as the reagent ion.
Isoprene 1N-, 2N-, and 3N-monomers and 2N- and 3N-dimers have different time behaviors, indicating the
occurrence of multi-generation oxidation during this process. Based on their specific time behaviors as well as
the general knowledge of isoprene and radical chemistry, the possible formation mechanisms of these
compounds are proposed.

In order to evaluate the potential contribution of various isoprene nitrates to SOA formation, different

composition-activity and group-contribution methods were used to estimate their saturation vapor pressures. We
also calculated the vapor pressures of isoprene oxidation products based on the gas-particle equilibrium
coefficients derived from condensation measurements. The vapor pressures estimated by different methods
spans several orders of magnitude, and the discrepancies increase as the compounds become highly
functionalized. It shows that existing group-contribution methods tend to underestimate the saturation vapor
pressure of the multifunctional low-volatility molecules, and we suggest that experimental methods might be a
good choice to estimate the volatility of highly oxidized compounds accurately.

According to our results, 1N-monomers and most 2N and 3N-nitrates fall in the IVOC or SVOC range.

Therefore, they have, with a few exceptions, low potential to form SOA at atmospheric organic aerosol loads. In
contrast, 2N- and 3N-dimers are estimated to have low or extremely low volatility, indicating that they are
significant contributors to SOA formation, although dimers constitute less than 2% of the total explained signals.
In this study, no new particle formation events were observed. Assuming that the dimers in the LVOC or
ELVOC range will condense onto particles completely, we estimate a SOA mass yield of about 5 %, which is a
lower limit if one takes a possible contribution of the minor dimer products as well as SVOC species into
consideration. Both the volatility distribution and calculated SOA yields indicate that isoprene dimers formed
from $NO_3$ oxidation are the major contributors to SOA formation.



**Data availability**

All data given in figures can be displayed in tables or in digital form. This includes the data given in the Supplement. The data will be made available via the repository Jülich DATA. Please send all requests for data to t.mentel@fz-juelich.de and r.wu@fz-juelich.de.

**Author contributions**

HF, JNC, JLF, SSB, AW, and AKS designed the study. Instrument deployment and data analysis were carried out by RW, ET, SK, SRA, LH, AN, HF, RT, TH, PTMC, JS, FB, BB, JAT. RW, LV, ET, DZ, JAT, MH, TFM interpreted the compiled data set. RW, TFM, LV wrote the manuscript. All co-authors discussed the results and commented on the manuscript.

**Competing interests**

The authors declare that they have no conflict of interest.

**Acknowledgements**

This work has received funding from the European Research Council (ERC) and European Commission (EC) under the European Union's Horizon 2020 research and innovation program (SARLEP grant agreement No. 681529, and Eurochamp 2020 grant agreement No. 730997). R.Wu gratefully acknowledges the fellowship from Helmholtz-OCPC (Office of China Postdoc Council) Postdoc Program for research support. M. Hallquist., Th.F.Mentel. and E.Tsiligiannis gratefully acknowledge the support by the Svenska Vetenskapsrådet (grant nos. 2014-05332 and 2018-04430) and the Svenska Forskningsrådet Formas (grant no. 2015-1537).

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
