# Peer review of "Molecular composition and volatility of multi-generation products formed from isoprene oxidation by nitrate radical"

_Atmospheric Chemistry and Physics, 2020_

## Referee Comment (RC1) · Anonymous Referee #1 · 8 Feb 2021

General comments

This study describes chamber studies of the chemistry of secondary organic aerosol (SOA) formation from the reactions of isoprene with nitrate radicals, illustrates the formation mechanisms of the multi-generation organic nitrates, investigates the volatilities of organic nitrates using both parametrization and experimental methods, and evaluates their potential to form SOA. The paper is well written, and the experiments and data analysis are well done. The mechanisms that are proposed are very plausible. Together with the volatility information of the organic nitrates, this study provides valuable information for understanding the isoprene-nitrate radical chemistry. I would

recommend the publication of the manuscript in Atmospheric Chemistry and Physics. However, there are a few points I would like the authors to clarify or add some more information.

Specific comments

Page 2 line 43-44: Do you mean organic nitrates monomers and organic nitrates dimers? Are CHO compounds also monomer and dimers?

Page 3 line 66-72: It is necessary to mention why isoprene, which is emitted in the daytime, also plays an important role in night-time chemistry.

Page 5 line 145-148: It is a little bit confusing how much you added for each injection "∼100, 30 and 10 ppbv of O3, NO2, and isoprene, respectively, were added... After another ∼ 1.5 hours, the chemistry was accelerated again by the third injection, and the concentrations of O3, NO2, and isoprene reached ∼ 100, 25, and 10 ppbv, respectively, after the injection." It is better to keep it consistent.

Page 5 line 150: Were there any differences among the different steps?

Page 6 line 174: How many compounds were identified and how many of them were deprotonated ions?

Page 7 line 180: Was PFPA used for mass calibration? How did you do the mass calibration for the range between 350 to 500+ Th?

Page 7 line 195: Did it happen only for one experiment or several during the campaign? It is not clear when you say "the influence from isomers and the differences in sensitivity between the two instruments."

Page 8 line 254: Was the nitrate group (-nN*bN) included in Donahue et al. (2011)?

Page 16 line 499-501: Could the reason also be that the Br- CIMS is not sensitive to those 4N and 5N-dimers, as they have many oxygen atoms?

Page 17 line 552-556: It would be interesting to compare this value to that of other OH, O3 initiated systems and/or the ambient aerosols. Could you make it clearer whether the increased OSc is due to the addition of nitrate group(s) only or other functional groups?

Page 26 line 831: The C* values from the three GC methods are higher than those from the experimental method.

Page 27 line 849: Similarly, the GC methods predict higher C* for less-functionalized. . .but lower C* for highly functionalized dimers.

Page 27 line 859-863: This study suggests that the experimental methods might give the most accurate volatility, but why did you use the median value of C* calculated by all parameterization and experimental methods for further evaluation of their potential to form SOA? The latter gave lower values of C* than that from only experimental methods, especially for those dimers with 2 and 3 nitrate groups. The SOA yield (5%) is estimated based on the assumption that all the dimers in the low- or extremely low-volatility range will condense completely. These compounds are exactly those dimers with 2 and 3 nitrate groups. Would the SOA yield become smaller by using values from the experimental methods? Please clarify this point and make it consistent.

Page 28 line 864: Why C5H9N3O12 has much higher C* than C5H9N3O10? If it is because of the same reason as you described for C5H9NO6 (vs. C5H9NO5), please also mention it.

Page 28 line 885-888: It would be good to clarify the RO2 fate (reactions with NO3, HO2, and RO2) for the whole experiment, as well as for different steps, and how it compared to the typical branching ratio in the atmosphere. It helps the comparison of the SOA yields to other studies and helps to interpret the multi-generation chemistry observed in this study in the ambient atmosphere.

Page 29 line 889: It is not clear that what are the "minor dimer products".

Page 29 line 915: What is the conclusion about the CA methods?

Technical corrections

Page 28 line 864: (1) Please put all markers on the top of the background (2) and change the legend (e.g. red markers for C5H8,10N2O8-10, and C5H6N2O8, C5H8N2O7, C4H6N2O7), please either use different colours, or change your legend.

Page 29 line 895: Please update the reference of Zhao et al.

SI Table S3: Please clarify which method was used for the volatility.

---

## Referee Comment (RC2) · Anonymous Referee #2 · 1 Apr 2021

MS No.: acp-2020-1180

The authors describe experimental results of the nitrate radical initiated oxidation of isoprene carried out in a batch-reactor for close to atmospheric conditions. Nitrate radical were generated via the O3 + NO2 pathway. The reaction was conducted in such a way that multiple nitrate radical attacks were possible leading to product formation of the 2nd and 3rd generation. Product formation was followed by a bromide-CIMS. The results are interesting and worth to be published in ACP. Some points should be considered before acceptance of the manuscript can be recommended.

1. Line 71. "isoprene nitrates"? better: "organic nitrates derived from isoprene oxidation"

2. Line 150, Fig.1a-c: It is stated that approx.90% of consumed isoprene reacted with NO3 based on modeling results. The reaction scheme with the used rate coefficients is not given yet. Please provide this information in the SI. The modeling results should be compared with the measurements of the chemical species depicted in Fig.1 (as done in a couple of other papers of the Jülich group). That gives the readership an impression how good the processes in the experiment have been understood and how accurate the model is.

3. Line 200-212: The authors determined a bulk sensitivity for the organic nitrates. It should be described more in detail what has been done. It is not clear to me why this calibration was not used to set all the measurements on an absolute scale. The authors argue "that the normalized signals are sufficient". More precise information is better in each case! On the other hand, they did it for C10 products. Why is the calibration only used for higher molecular products? Please comment. It would be fine if the authors could provide the plots of the C5 org.nitrate concentrations (or the yields because the amount of reacted isoprene is known). And please add a discussion regarding the uncertainty of these absolute values.

4. Line 417, Fig.2: In Fig.2 an average spectrum from the complete experiment is given. It would be fine having also a product spectrum from the first injection showing mainly 1st generation products. A couple of possible reaction pathways were mentioned/discussed in the paragraph before, incl. possible RO2 isomerization step leading possibly to HOMs. Nothing is said here regarding the relevance of RO2 isomerization in this reaction system based on the measurements. It would be also very helpful for the readership to have a reaction scheme in the main body that explains the formation of the observed main products, i.e. C5H9NO5, C4H7NO5, C5H9NO4, C5H9NO6, .... What about the formation of the carbonyls HCHO, MVK, MACR? The Vocus PTR-MS is very sensitive at least for the C4-carbonyls.

---

## Author Comment (AC1) · 24 May 2021

**Response Letter to Referee #1**

The authors thank the reviewer for careful reading and positive feedbacks. We also appreciate the reviewer for the helpful comments and suggestions, which significantly improved our manuscript. All the comments are addressed below point by point in bold text, with our response followed in non-bold text and the corresponding revisions to the manuscript in blue. All updates to the original submission were tracked in the revised version as you can find below.

**Anonymous Referee #1:**

**General comments**

**This study describes chamber studies of the chemistry of secondary organic aerosol (SOA) formation from the reactions of isoprene with nitrate radicals, illustrates the formation mechanisms of the multi-generation organic nitrates, investigates the volatilities of organic nitrates using both parameterization and experimental methods, and evaluates their potential to form SOA. The paper is well written, and the experiments and data analysis are well done. The mechanisms that are proposed are very plausible. Together with the volatility information of the organic nitrates, this study provides valuable information for understanding the isoprene-nitrate radical chemistry. I would recommend the publication of the manuscript in Atmospheric Chemistry and Physics. However, there are a few points I would like the authors to clarify or add some more information.**

**Specific comments**

**1. Page 2 line 43-44: Do you mean organic nitrates monomers and organic nitrates dimers? Are CHO compounds also monomer and dimers?**

**Response:** Yes, monomers and dimers in line 43-44 in the original manuscript refer to organic nitrate monomers and organic nitrate dimers. According to the modeling results, approximately 90% of the isoprene reacted with $NO_3$, and most of the corresponding products were organic nitrates. There were few CHO compounds formed in isoprene-$NO_3$ system, and their signal

intensities were relatively low. The identified CHO compounds were $C_5$- or even smaller molecules, no CHO dimers were detected.

To clarify, the original sentence "They are grouped into monomers ($C_4$- and $C_5$-products), and dimers ($C_{10}$-products) with 1–3 nitrate groups according to their chemical composition" is revised to "Most of the products detected are organic nitrates, and they are grouped into monomers ($C_4$- and $C_5$-products), and dimers ($C_{10}$-products) with 1–3 nitrate groups according to their chemical composition".

**2. Page 3 line 66-72: It is necessary to mention why isoprene, which is emitted in daytime, also plays an important role in night-time chemistry.**

**Response:** We thank the reviewer for pointing this out. We added some sentences in the revised manuscript to clarify this.

In the revised manuscript, the original sentences "At night when the concentration of OH is negligible, the nitrate radical ($NO_3$) and $O_3$ become the predominant oxidants of isoprene. Reactions of isoprene with $NO_3$ is competitive to that with $O_3$ because …lower than that of $O_3$." is revised to "Although the majority of isoprene emissions is emitted by plants and is light-dependent, isoprene emitted in the day can persist in the boundary layer after sunset, and its mixing ratio can remain as high as several ppb (Brown et al., 2009; Starn et al., 1998; Stroud et al., 2002; Warneke et al., 2004). During the daytime, isoprene is primarily oxidized by the hydroxyl radical (OH) and somewhat by ozone ($O_3$), but its main oxidizers shift to nitrate radical ($NO_3$) and $O_3$ in the nighttime (Wennberg et al., 2018). Due to the higher reactivity of $NO_3$ with isoprene ($k_{NO_3}$ = 6.5 $\times 10^{-13}$ cm$^3$ molecules$^{-1}$s$^{-1}$ and $k_{O_3}$ = 1.28 $\times 10^{-17}$ cm$^3$ molecules$^{-1}$s$^{-1}$ at 298 K, respectively, IUPAC), a considerable fraction of the residual isoprene would be oxidized by $NO_3$ at night, and therefore nocturnal nitrate radical chemistry is typically thought to be of significant importance for isoprene, especially in regions where sufficient nitrogen oxides are available (Brown et al., 2009; Fry et al., 2018; Ng et al., 2017; Wennberg et al., 2018)."

**3. Page 5 line 145-148: It is a little bit confusing how much you added for each injection "~ 100, 30 and 10 ppbv of O3, NO2, and isoprene, respectively, were added… After another ~ 1.5 hours, the chemistry was accelerated again by the third injection, and the concentrations**

**of O3, NO2, and isoprene reached ~ 100, 25, and 10 ppbv, respectively, after the injection."**
**It is better to keep it consistent.**

**Response:** We thank the reviewer for the comment. In the revised manuscript, we reorganized the original sentences as follows: "The second injection was done when isoprene from the first injection was almost completely consumed, to reach concentrations of $O_3$, $NO_2$, and isoprene in the chamber of ~ 100, 30, and 10 ppbv, respectively. About 1.5 hours later, the chemistry was further accelerated by a third injection of precursors, and accordingly the concentrations of $O_3$, $NO_2$, and isoprene in the chamber reached ~ 100, 25, and 10 ppbv, respectively."

**4. Page 5 line 150: Were there any differences among the different steps?**

**Response:** Only minor changes, as shown in Fig. S1 below, there were few differences in the relative fraction of isoprene consumed by $O_3$ and $NO_3$ among different steps. In addition to the newly added plot in the revised SI, we added some discussion (in lines 164-168) in the revised manuscript to explain why isoprene loss due to reaction with OH was not considered in our system.

In our system (under dark condition), OH is produced from isoprene ozonolysis (Nguyen et al., 2016), but its concentration was below the detection limit of the instrument during our experiment (see Fig. S2). Therefore, isoprene loss due to reaction with OH could not be quantified from the measurement. However, OH is expected to contribute about 10% of the isoprene losses, with the contribution of the $NO_3$ reaction accounting for up to 80%, as determined in a recently published modelling work based on the same campaign, which implemented a newly developed $NO_3$-isoprene mechanism with updated $RO_2$ and RO chemistry (Vereecken et al., 2021).

[Figure]

**Figure S1: Fraction of isoprene consumed by O₃ and NO₃ calculated from measurements. The amount of isoprene consumed by O₃ was calculated from the measured concentrations of O₃ and isoprene, and that consumed by NO₃ was calculated from total isoprene losses (from isoprene measurements) subtracting losses due to dilution and reaction with O₃. Isoprene losses due to reaction with OH could not be included here, because the OH concentration was below the detection limit. However, according to model calculations OH contributed about 10% to the isoprene consumption (Vereecken et al., 2021).**

**5. Page 6 line 174: How many compounds were identified and how many of them were deprotonated ions?**

**Response:** In total, about 190 ions were identified for each mass spectrum of which more than 80% were detected as adducts with $Br^-$. In addition, ~ 7% of the identified species were supposed to be deprotonated ions, while another 10% were identified as adducts with $NO_3^-$. We have added these details in the revised manuscript.

In the revised manuscript, the original sentence "In our system, most compounds were detected as adducts with $Br^-$, but some strong acidic compounds like nitric acid were also detected as deprotonated ions." is replaced by "In our system, on average, about 190 ions were identified for each mass spectrum, most of which were detected as adducts with $Br^-$, while some acidic compounds (~ 7% of the total) like nitric acid ($HNO_3$), glycolic acid ($C_2H_4O_3$), and malonic acid ($C_3H_4O_4$) were also detected as deprotonated ions. In addition, there were some ions (~ 10% of the total) identified as adducts with $NO_3^-$."

**6. Page 7 line 180: Was PFPA used for mass calibration? How did you do the mass calibration for the range between 350 to 500+ Th?**

**Response:** Yes, PFPA was used for mass-to-charge ratio ($m/z$) calibration. In total, five isolated peaks were used, including $Br^-$, $H_2OBr^-$, $HNO_3Br^-$, $C_5F_9O_2^-$, and $C_5F_9HO_2Br^-$, as shown below, over the mass range of dominant products (200-350 Th). Due to its low signal intensity, PFPA cluster ($C_{10}F_{18}O_4H^-$, $m/z$ 526.959290) was not defined as a calibrant, and there were no other suitable masses with sufficient intensity and high accuracy that can be used to calibrate the higher mass range. Therefore, it should be noted that the peak fitting in the mass range between 300 to 500 Th or even above might have higher uncertainties.

**Table: Isolated peaks used for mass-to-charge ratio calibration**

| Exact $m/z$ | Ion formula | Accuracy $\pm$ 1$\sigma$ (ppm) |
|---|---|---|
| 78.918886 | $Br^-$ | 4 $\pm$ 2 |
| 96.929450 | $H_2OBr^-$ | 3 $\pm$ 1 |
| 141.914529 | $HNO_3Br^-$ | 5 $\pm$ 4 |
| 262.976007 | $C_5F_9O_2^-$ | 5 $\pm$ 4 |
| 342.902169 | $C_5F_9HO_2Br^-$ | 2 $\pm$ 2 |

In the revised manuscript, we added discussion of this point. "For $m/z$ calibration, five isolated peaks were used, including $Br^-$ ($m/z$ 79), $H_2OBr^-$ ($m/z$ 97), $HNO_3Br^-$ ($m/z$ 142), $C_5F_9O_2^-$ ($m/z$ 263), and $C_5F_9HO_2Br^-$ ($m/z$ 343), covering the mass range of dominant products. The averaged accuracies of all five calibrated masses were below 5 ppm over the whole measurement period. However, due to the low signal intensity, the PFPA cluster ($C_{10}F_{18}O_4H^-$, $m/z$ 527) was not suited for mass calibration, and there were no other suitable masses with sufficient intensity and high accuracy that could be used to calibrate the higher mass range. Therefore, peak fitting in the mass range between 300 to 500+ Th might have higher uncertainties." is added in line 201-207 in the revised manuscript.

**7. Page 7 line 195: Did it happen only for one experiment or several during the campaign? It is not clear when you say "the influence from isomers and the differences in sensitivity between the two instruments."**

**Response:** It was observed for several experiments during the campaign, and for different compounds. The correlation coefficients of measurements for compounds with the same chemical formula from two instruments (bromide and iodide CIMS) deviated from experiment to experiment. This is probably related to different experimental conditions, which might lead to different chemical processes and thus formation of isomers. Since CIMS might have different sensitivities to isomers, and different instruments with different ionization schemes are selective for different compounds, it is explicable the correlation coefficients of measurements from Br⁻ and I⁻ CIMS differed from day to day.

In the revised manuscript, the original sentence was rewritten and more discussion was added to make it clearer: "As shown in Fig. S4b, … . However, the correlation coefficients of measurements from two instruments deviated from experiment to experiment. This is probably related to different experimental conditions, which might lead to different chemical processes and thus formation of isomers. Since CIMS with different reagent ions might have different sensitivities to isomers, and may be selective for different compounds, correlation coefficients of measurements from Br⁻ and I⁻ CIMS may differ from day to day. Moreover, the Br⁻-CIMS was not tuned during the campaign, while the I⁻ CIMS was optimized from time to time."

**8. Page 8 line 254: Was the nitrate group (-Nn*bN) included in Donahue et all. (2011)?**

**Response:** Yes, the effect of the presence of nitrogen on vapor-pressure estimation was considered for method by Donahue et al. (2011) by assuming that all nitrogen atoms in the detected compounds are nitrate functional groups (–ONO$_2$) and each group would lower C* by 2.5 orders of magnitudes. The vapor pressure of a given compound was calculated by following four-parameter expression:

$$log_{10}C^* = (n_C^0 - n_C)b_C - n_O b_O - 2\frac{n_C n_O}{n_C + n_O}b_{CO} - n_N b_N$$

where $n_C^0 = 25$, $b_C = 0.475$, $b_O = 0.2$, $b_{CO} = 0.9$, $b_N = 2.5$. $n_C$, $n_O$, and $n_N$ are the numbers of carbon, oxygen, and nitrogen atoms in the compound.

In the revised manuscript, we emphasize this point by adding "All of these three parameterization methods have included the effect of the presence of nitrate groups on vapor-pressure estimation." right after the original sentence "These include parameterizations that were

constrained by chamber measurements as proposed by Donahue et al. (2011), Mohr et al. (2019), and Peräkylä et al. (2019)."

**9. Page 16 line 499-501: Could the reason also be that the Br-CIMS is not sensitive to those 4N and 5N-dimers, as they have many oxygen atoms?**

**Response:** We thank the reviewer for raising this point, but we think there is no evidence that the $Br^-$ ionization efficiency would be lower to the 4N- and 5N-dimers due to their high oxygen content. We rather believe that the $Br^-$ ionization efficiency is more likely as high or higher for higher oxygen-containing species, similar to $I^-$.

In the revised manuscript, we included this point by revising the original sentence to "However, … , suggesting that the 4N- and 5N-dimers were either not formed, or if present, with lower absolute concentrations below the instrument detection limit (approximately $5\times10^7$ and $5\times10^5$ molecules $cm^{-3}$ for salicylic acid and acetic acid, for an integration time of 60 s)."

**10. Page 17 line 552-556: It would be interesting to compare this value to that of other OH, O3 initiated systems and/ or the ambient aerosols. Could you make it clearer whether the increased OSc is due to the addition of nitrate group(s) only or other functional groups?**

**Response:** We thank the reviewer for this comment. In the revised manuscript, we added a paragraph that compares the average carbon oxidation state ($\overline{OS_C}$) in this study to that of OH and O3 initiated system, as well as to that of ambient organic aerosol. In addition, we added discussion about contributors to the increased $\overline{OS_C}$.

The value of $\overline{OS_C}$ is determined by the relative abundances and oxidation states of atoms in the compound, which could vary from system to system due to different chemical compositions and oxidation conditions. In our study, the $\overline{OS_C}$ of products from NO3-initiated isoprene oxidation system ranged from -0.35 to 0.09 depending on the degree of oxidation. For OH- and O3-intiated systems, the average oxidation state of laboratory-generated isoprene SOA ranged from -1.3 to -0.2, as listed in Table S4. It seems that the SOA generated from chloride-initiated oxidation of isoprene is more oxidized compared to other isoprene oxidation systems, of which the $\overline{OS_C}$ can be as high as +1.8 (Wang and Ruiz, 2017). With regard to ambient measurements, the calculated $\overline{OS_C}$ values of organic aerosol and aerosol fractions fell into a wider range between -2 to +2, depending on the site position and the corresponding oxidation environment of that site (Table S4).

In the revised manuscript, we added following discussion which compares the $\overline{OS_C}$ of different isoprene oxidation systems and ambient organic aerosols: "As mentioned above, the average carbon oxidation state of a mixture of molecules largely depends on its chemical composition. Therefore, for different oxidation systems, their $\overline{OS_C}$ may differ due to different precursors and oxidation conditions. In our study, the $\overline{OS_C}$ of $NO_3$-initiated isoprene oxidation system increased from -0.35 to 0.09 with further oxidation. For OH- and $O_3$-initiated systems, the average oxidation state of laboratory-generated isoprene SOA are reported to range from -1.3 to -0.2, as listed in Table S4. It seems that the SOA generated from chloride-initiated oxidation of isoprene is more oxidized compared to other isoprene oxidation systems, for which the $\overline{OS_C}$ can be as high as +1.8 according to limited studies (Wang and Ruiz, 2017). With regard to ambient measurements, the calculated $\overline{OS_C}$ values of organic aerosol and aerosol fractions fell into a wider range between -2 to +2, depending on the site position and the corresponding oxidation environment of that site (Table S4)."

In the oxidation system, the increase in $\overline{OS_C}$ is attributed to the formation of bonds between carbon and oxygen as well as other electronegative atoms, and/ or the breaking of bonds between carbon and hydrogen and other electropositive atoms (Kroll et al., 2011). The oxidation has an inherent directionality, for which the starting point is the precursor (here isoprene, and thus $OS_C =$ -1.6) and its ultimate ending point is $CO_2$ ($OS_C = + 4$). The nitrate functional group (–$ONO_2$) has a group oxidation state of -1, which means that addition of a –$ONO_2$ group to isoprene will increase its $OS_C$ by 0.2. According to our estimation, the values of system $\overline{OS_C}$ increased by 1.25 (step I), 0.09 (step II), and 0.35 (step IV), respectively, indicating that the increases in $\overline{OS_C}$ are not only due to addition of –$ONO_2$ group(s) but also other oxygen-containing functionalities. In addition to functionalization, it's possible that other reactions such as fragmentation and oligomerization which can increase or reduce the oxidation state were involved during the reaction.

In the revised manuscript, we added following discussion as a separate paragraph to address the above point: "In the oxidation system, the increase in $\overline{OS_C}$ is attributed to the formation of bonds between carbon and oxygen as well as other electronegative atoms, and/ or the breaking of bonds between carbon and hydrogen and other electropositive atoms (Kroll et al., 2011). The –$ONO_2$ group has an oxidation state of -1, which means that addition of a –$ONO_2$ group to isoprene will increase its $OS_C$ by 0.2. According to our estimates, the values of system $\overline{OS_C}$ increased by 1.25 (step I), 0.09 (step II), and 0.35 (step IV), indicating that the increases in $\overline{OS_C}$ are not only

due to addition of –ONO₂ group(s) but also to other oxygen-containing functionalities. In addition to functionalization, it is possible that other reactions such as fragmentation and oligomerization which can increase or reduce the oxidation state were involved during the reaction.".

**Table S4: Average carbon oxidation state ($\overline{OS_C}$) measured from ambient or different isoprene oxidation systems**

| | $\overline{OS_C}$ | Technique | Ref. |
|---|---|---|---|
| **Chamber measurements** | | | |
| 1. **Gas-phase products** | | | |
| Isoprene + NO₃ | -0.35 to + 0.09 | Br⁻ CIMS | This study |
| Isoprene 4-hydroxy-3-hydroperoxy (4,3-ISOPOOH) + OH | avg. +0.10 | NO₃⁻ CIMS | Krechmer et al., 2015 |
| 2. **Secondary organic aerosol** | | | |
| Isoprene + OH | ~ -1.3 | AMS | Aiken et al., 2008 |
| Isoprene + O₃ | -0.31 (+)/ -0.25 (-) [a] | ESI-MS | Nguyen et al., 2010 |
| Isoprene + OH | -0.53 to -0.35 | AMS | Chhabra et al., 2010 |
| 4,3-ISOPOOH + OH | avg. +0.05 | AMS | Krechmer et al., 2015 |
| Isoprene + OH | avg. -0.7 | AMS | Lambe et al., 2015 |
| Isoprene + Cl | -0.5 to +1.8 | ACSM | Wang and Ruiz, 2017 |
| **Ambient measurements** | | | |
| 1. **Ambient organic aerosol** | | | |
| Mexico City, US | -1.54 to +0.11 | AMS | Aiken et al., 2008 |
| Amazonia | -0.9 to -0.2 | AMS | Chen et al., 2009 |
| Whistler Peak, Canada | avg. -0.14 | AMS | Sun et al., 2009 |
| Kaiping, China | avg. -0.54 | AMS | Huang et al., 2011 |
| Melpitz, Germany | avg. -0.47(su)/ -0.4(a)/ -0.41(w) [b] | AMS | Poulain et al., (2011) |
| Hongkong, China | avg. -0.59 (sp)/ -0.32 (su)/ -0.55 (a)/ -0.54 (w) [b] | AMS | Li et al., 2015 |
| Mount Wuzhi, China | avg. +0.64 | AMS | Zhu et al., 2016 |
| Lake Hongze, China | avg. -0.18 | AMS | Zhu et al., 2016 |
| Beijing, China | avg. -0.64 (sp)/ -0.54 (su)/ -0.66 (a)/ -0.58 (w) [b] | AMS | Hu et al., 2017 |
| Houston, US | avg. -0.09 | AMS | Al-Naiema et al., 2018 |
| Seoul, Korea | avg. -0.80 | AMS | Kim et al., 2018 |
| Zurich, Switzerland | avg. -0.31 (w)/ -0.25 (su) [b] | ESI-UHR-MS | Daellenbach et al., 2019 |

| | | | |
|---|---|---|---|
| Hyytiälä, Finland | avg. -0.66 (2011)/ -0.36 (2014) [c] | ESI-UHR-MS | Daellenbach et al., 2019 |
| Waliguan, China | avg. +0.55 | AMS | Zhang et al., 2019 |
| Yorkville, US | avg. +0.15 | FIGAERO-CIMS | Chen et al., 2020 |
| Yorkville, US | avg. -0.12 | AMS | Chen et al., 2020 |
| Xinglong, China | avg. -0.01 (sp)/ + 0.10 (su)/ 0.26 (a)/ -0.45 (w) [b] | AMS | Li et al., 2021 |
| Oklahoma, US | avg. +0.289 (sp)/ -0.34 (su) [b] | AMS | Liu et al., 2021 |
| **2. Ambient aerosol fractions** | | | |
| Hydrocarbon-like OA (HOA) | -1.7 to -1.6 | AMS | Aiken et al., 2008 |
| Oxygenated OA type I (OOA-I) | -0.5 to 0.0 | AMS | Aiken et al., 2008 |
| Oxygenated OA type II (OOA-II) | + 0.5 to +0.9 | AMS | Aiken et al., 2008 |
| Daytime more oxidized OA | avg. +0.50 | FIGAERO-CIMS | Chen et al., 2020 |
| Daytime ON-rich OA | avg. +0.35 | FIGAERO-CIMS | Chen et al., 2020 |
| Morning less oxidized OA | avg. -0.13 | FIGAERO-CIMS | Chen et al., 2020 |
| Afternoon less oxidized OA | avg. +0.04 | FIGAERO-CIMS | Chen et al., 2020 |
| Nighttime ON-rich OA | avg. +0.13 | FIGAERO-CIMS | Chen et al., 2020 |
| OOA-I | avg. +0.14 (sp)/ -0.099 (su) [b] | AMS | Liu et al., 2021 |
| OOA-II | avg. -0.315 (sp)/ -0.264 (su) [b] | AMS | Liu et al., 2021 |
| Isoprene-epoxydiol-derived SOA (IEPOX SOA) | avg. +1.606 (sp)/ -0.096 (su) [b] | AMS | Liu et al., 2021 |
| HOA | avg. -1.91 | AMS | Al-Naiema et al., 2018 |
| Cooking-influenced less-oxidized oxygenated OA | avg. -0.35 | AMS | Al-Naiema et al., 2018 |
| More-oxidized oxygenated OA | avg. +1.27 | AMS | Al-Naiema et al., 2018 |
| **3. Gas-phase products** | | | |
| Centreville, US | avg. -0.043 | $NO_3^-$ CIMS | Massoli et al., 2018 |

[a] Values calculated separately for data from the positive (+) and negative (-) ion mode mass spectra
[b] Values calculated separately for data collected in spring (sp), summer (su), autumn (a) and winter (w).
[c] Values calculated for data collected in 2011 and 2014

**11. Page 26 line 831: The C\* values from the GC methods are higher than those from the experimental method.**

**Response:** In the revised manuscript, we corrected this error.

**12. Page 27 line 849: Similarly, the GC methods predict higher C\* for less-functionalized… but lower C\* for highly functionalized dimers.**

**Response:** In the revised manuscript, we corrected this error.

**13. Page 27 line 859-863: This study suggests that the experimental methods might give the most accurate volatility, but why did you use the median value of C\* calculated by all parameterization and experimental methods for further evaluation of their potential to form SOA? The latter gave lower values of C\* than that from only experimental methods, especially for those dimers with 2 and 3 nitrate groups. The SOA yield (5%) is estimated based on the assumption that all the dimers in the low- or extremely low-volatility range will condense completely. These compounds are exactly those dimers with 2 and 3 nitrate groups. Would the SOA yield become smaller by using values from the experimental methods? Please clarify this point and make it consistent.**

**Response:** According to the results from different vapor-pressure estimation methods and our knowledge of each method, it seems that the experimental method provides more accurate estimations compared to other parameterization methods, but there is or are limitation(s) for the experimental method. The gas-to-particle partitioning is a complex and dynamic equilibrium, and it might induce multi-phase and/or heterogeneous reactions. For example, some of the detected compounds had higher signals in the experiment with seed aerosol than that without seed aerosol, which is an unexpected observation. Our explanation is that this can probably be attributed to heterogenous reactions as discussed in the main body in sect. 3.3.1, and under this condition it fails to calculate vapor-pressure using the experimental method we proposed in this study. As a consequence, there are no estimates for these compounds from the experimental method, which thus limits its usage. Therefore, we finally decided to use the median value of $C^*$ calculated by different methods as the estimator of the vapor pressure of each compound.

Since there are only four estimated $C^*$ values from the experimental methods, including that for $C_{10}H_{16}N_2O_{11}$, $C_{10}H_{16}N_2O_{13}$, $C_{10}H_{17}N_3O_{12}$, and $C_{10}H_{17}N_3O_{13}$, we took these four compounds as examples to evaluate the effect of using experimental results instead of median values of results from all methods adopted in this study on the estimated SOA yield. According to the results from experimental methods, all of above four compounds belong to SVOC range. Assuming that all of

the original 2N- and 3N-dimers that belong to LVOC or ELVOC range turn to SVOC range if using experimentally determined vapor pressures, the fraction of 2N- and 3N-dimers in particle phase are estimated to be ~ 0.3 and 0.7 based on the gas-to-particle partitioning equilibrium theory (with their average saturation concentrations being 4 $\mu g\ m^{-3}$ and 1 $\mu g\ m^{-3}$, respectively, and the average organic aerosol loading of 2 $\mu g\ m^{-3}$ as measured by AMS in the experiment with seeds). In total, the average SOA mass yield from dimer condensation is estimated to be 3.4%, a little bit smaller than the original value (5 % $\pm$ 2 %), but in the uncertainty range.

**14. Page 28 line 864: Why C5H9NO12 has much higher C\* than C5H9NO10? If it is because of the same reason as you described for C5H9NO6 (vs. C5H9NO5), please also mention it.**

**Response:** Yes, it is assumed to be due to the same reason. As listed in Table S2, $C_5H_9N_3O_{10}$ is taken to be a hydroxyl compound, while for $C_5H_9N_3O_{12}$ there are two possibilities. One is proposed to be hydroperoxy peroxynitrate, and the other to be hydroxy hydroperoxyl compounds. Since molecules with multiple polar functional groups like hydroperoxyl, peroxy acid, and peroxide functionalities are prone to form intramolecular H-bonding, which can substantially increase the vapor pressure (Bilde et al., 2015; Kurten et al., 2016), $C_5H_9N_3O_{12}$ is expected to have a higher vapor pressure than $C_5H_9N_3O_{10}$.

In the revised manuscript, we added the following sentence "This explanation is also valid for $C_5H_9N_3O_{10}$ and $C_5H_9N_3O_{12}$." right after the discussion on $C_5H_9NO_5$ and $C_5H_9NO_6$.

**15. Page 28 line 885-888: It would be good to clarify the RO2 fate (reactions with NO3, HO2, and RO2) for the whole experiment, as well as for different steps, and how it compared to the typical branching ratio in the atmosphere. It helps the comparison of the SOA yields to other studies and helps to interpret the multi-generation chemistry observed in this study in the ambient atmosphere.**

**Response:** We thank the reviewer for pointing out this.

In a recently published work from Vereecken et al. (2021) based on the same campaign, the fates of $RO_2$ derived from $NO_3$-initiated oxidation of isoprene under different chamber conditions have been investigated by modeling, with using updated unimolecular reaction rate coefficients from theoretical calculations. The results show that reactions with $HO_2$ and $NO_3$ were the dominant loss channels for the $RO_2$ formed from $NO_3$-initiated oxidation of isoprene under our chamber

conditions. $RO_2 + RO_2$ reactions were never dominant, even in the experiment with the highest modelled peak $RO_2$ concentrations. For some specific $RO_2$, unimolecular reactions became the dominant loss channel due to their fast reaction rate coefficients, albeit its overall contribution was still small (Vereecken et al., 2021). Following the modelling results from Vereecken et al. (2021), Brownwood et al. (2021) provided a breakdown of the fates of $RO_2$ for four different experimental days. Here, we have listed the results for 13 August from Brownwood et al. (2021), which was at almost the same conditions as 08 August used for analysis in our study. Overall, reaction with $HO_2$ and $NO_3$ contributed for ~ 53% and ~ 30% of total $RO_2$ loss, followed by $RO_2 + RO_2$ reaction (~ 13%) and unimolecular reaction (~ 5%). There are few differences among the first three injection periods, while the sustained $RO_2$ concentration during the fourth injection period remained negligible, as there was no isoprene added. Since the experimental conditions of 08 August were almost identical to those of 13 August, we expect that the majority of the $RO_2$ were lost by reaction with $HO_2$ and $NO_3$ in our experiment as well, with $RO_2 + RO_2$ only being a minor loss channel.

In polluted urban regions, nocturnal $NO_3$ concentration can reach to as high as several hundreds of ppt (Brown and Stutz, 2012), and the fate of $RO_2$ is typically dominated by $RO_2 + NO_3$ (Boyd et al., 2015). In comparison, in the more pristine environment, where $HO_2$ concentration is typically high while $NO_3$ concentration is low, the $RO_2 + HO_2$ reaction will dominate $RO_2$ fate (Bianchi et al., 2019; Boyd et al., 2015).

$RO_2 + HO_2$ was more important in the chamber than that in ambient and enhanced $RO_2 + HO_2$ would potentially lead to less dimer formation by $RO_2 + RO_2$ reactions and hence reduce SOA yields. However, recent work from Brownwood et al. (2021) based on the same campaign pointed out that the bulk aerosol composition and SOA yields were largely independent of $RO_2$ fate. In addition, Boyd et al. (2015) found that SOA yields in the "$RO_2 + NO_3$ dominant" and "$RO_2 + HO_2$ dominant" experiments were comparable for the β-pinene-$NO_3$ system. Consequently, the SOA yield estimated in this study is likely to be comparable to that in the atmosphere.

In the revised manuscript, we added discussion about the fate of $RO_2$ and compared with that in the atmosphere as follows:

"The fate of $RO_2$ determines the product distribution directly and hence could substantially affect SOA yields and aerosol physicochemical properties (Boyd et al., 2015; Fry et al., 2018; Ng et al., 2008; Schwantes et al., 2015; Ziemann and Atkinson, 2012). Consequently, it would be helpful to provide SOA yields together with the fate of $RO_2$. In our experiment, reactions with

HO$_2$ and NO$_3$ were the dominant loss channels for the initially formed RO$_2$ from isoprene oxidation by NO$_3$, contributing for ~ 53% and ~ 30% of overall RO$_2$ loss; RO$_2$ + RO$_2$ reactions contributed a minor fraction (~ 13%) followed by unimolecular reactions with a contribution of ~ 5%, according to modelling results (Brownwood et al., 2021). More details about the modelling and the results can be found elsewhere (Brownwood et al., 2021; Vereecken et al., 2021).

In polluted urban regions, the fate of RO$_2$ is typically dominated by RO$_2$ + NO$_3$, while in the more pristine environment, the RO$_2$ + HO$_2$ reaction will dominate RO$_2$ fate (Bianchi et al., 2019; Boyd et al., 2015; Brown and Stutz, 2012). RO$_2$ + HO$_2$ was more important in the chamber than that in ambient and enhanced RO$_2$ + HO$_2$ would potentially lead to less dimer formation by RO$_2$ + RO$_2$ reactions and hence reducing SOA yields. However, a recent work from Brownwood et al. (2021) based on the same campaign as this study pointed out that the bulk aerosol composition and SOA yields were largely independent of RO$_2$ fate. Similarly, Boyd et al. (2015) found for β-pinene-NO$_3$ system that RO$_2$ fate ("RO$_2$ + NO$_3$ dominant" vs "RO$_2$ + HO$_2$ dominant") had only few effects on SOA formation. Therefore, the SOA yield estimated in this study is expected to be comparable to that in the atmosphere."

**Table S3: Breakdown of the reactivity fate of the initially formed nitrate peroxy radicals for 13 August, nomenclature and mechanism following Vereecken et al., 2021. Relative weight of 1-nitrate to 4-nitrate addition sites are modeled as 87/13. The equilibrium ratio of the site specific peroxy radicals was calculated to be 8% Z-ISOP1N4OO, 18% *E*-ISOP1N4OO, 74% ISOP1N2OO and 20% Z-ISOP1OO4N, 40% *E*-ISOP1OO4N, 40% ISOP3OO4N, respectively. Only the Z-conformers have contributing unimolecular decomposition pathways.**

| 13 August "RO$_2$ enhanced" (faster RO$_2$ production) | unimolecular loss / % | reaction with HO$_2$ / % | reaction with NO$_3$ / % | reaction with RO$_2$ / % |
|---|---|---|---|---|
| *Z*-ISOP1N4OO | 45 | 20 | 11 | 24 |
| *E*-ISOP1N4OO | - | 34 | 19 | 47 |
| ISOP1N2OO | - | 64 | 36 | 1 |
| **all 1-nitrate RO$_2$** | **4** | **55** | **31** | **11** |
| *Z*-ISOP1OO4N | 74 | 11 | 7 | 8 |
| *E*-ISOP1OO4N | - | 38 | 22 | 40 |
| ISOP3OO4N | - | 53 | 31 | 17 |
| **all 4-nitrate RO$_2$** | **15** | **38** | **22** | **25** |
| **all nitrate RO$_2$** | **5** | **53** | **30** | **13** |

**16. Page 29 line 889: It is not clear that what are the "minor dimer products".**

**Response:** Sorry for being unclear. As mentioned at the beginning of this graph, we estimated the SOA yield for condensation with the assumption that dimers in the LVOC or ELVOC range would completely partition into the particle phase. Dimers identified in this study are supposed to be LVOC or ELVOC except for $C_{10}H_{16}N_2O_8$ and $C_{10}H_{16}N_2O_8$, which are in SVOC range (as shown in Fig. 8). Here the "minor dimer products" refers to $C_{10}H_{16}N_2O_8$ and $C_{10}H_{16}N_2O_9$.

We replaced the original sentence "The SOA yield will probably become somewhat higher if taking the contribution of the minor dimer products as well as SVOCs into consideration." by "The SOA yield will probably become somewhat higher if taking the contribution of SVOCs (including $C_{10}H_{16}N_2O_8$, $C_{10}H_{16}N_2O_9$ and some other monomers, as shown in Fig. 8) into consideration" in the revised manuscript.

**17. Page 29 line 915: What is the conclusion about the CA methods?**

**Response:** The estimates from composition-activity methods, especially those from Donahue et al. method, seriously underestimated the results from experimental methods.

In the revised manuscript, we added this point by reorganizing the original sentence to "It shows that existing composition-activity methods (especially the Donahue et al. method) seriously underestimate the saturation vapor pressure of multifunctional low-volatility molecules compared to the experimental methods. The group-contribution methods seem to have a better performance than the CA methods on this aspect, but they still have a tendency for to underestimation. We suggest that experimental method is a good choice to estimate the volatility of highly oxidized compounds accurately."

**Technical corrections**

**18. Page 28 line 864: (1) Please put all markers on the top of the background (2) and change the legend (e.g., red markers for C5H8,10N2O8-10, and C5H6N2O8, C5H8N2O7, C4H6N2O7), please either use different colours, or change you legend.**

**Response:** The redrawn plot is shown as follows:

[Figure]

**Figure 8: Volatility distribution of different organonitrates formed from NO₃-initiated isoprene oxidation. The volatility classes are indicated along the top with corresponding colors in the plot. The position of potential SOA contributors is determined depending on the exact functionalities of molecules adapted from Bianchi et al. (2019).**

**19. Page 29 line 895: Pease update the reference of Zhao et al.**

**Response:** Done.

**20. SI Table S3: Please clarify which method was used for the volatility.**

**Response:** In the revised manuscript, a note is added below Table S3 to clarify the method used for volatility estimation as follows:

**Table S3: Estimated wall loss rates of different dimers**

| Formula | Volatility range [a] | Used for yield calculation? | $k_w$ (s⁻¹) | $\tau_w$ (s) | $\tau_{dil}$ (s) |
|---|---|---|---|---|---|
| $C_{10}H_{16}N_2O_8$ | SVOC | No | 1.59E-05 | 8.67E+04 | 7.2E+04 |
| $C_{10}H_{16}N_2O_9$ | SVOC | No | 2.99E-05 | 4.62E+04 | 7.2E+04 |
| $C_{10}H_{16}N_2O_{10}$ | LVOC | Yes | 4.57E-05 | 2.82E+04 | 7.2E+04 |

| | | | | | |
|---|---|---|---|---|---|
| $C_{10}H_{16}N_2O_{11}$ | LVOC | Yes | 7.76E-05 | 1.31E+04 | 7.2E+04 |
| $C_{10}H_{16}N_2O_{12}$ | LVOC | Yes | 1.12E-04 | 8.30E+03 | 7.2E+04 |
| $C_{10}H_{16}N_2O_{13}$ | ELVOC | Yes | 2.02E-04 | 3.18E+03 | 7.2E+04 |
| $C_{10}H_{17}N_3O_{12}$ | LVOC | Yes | 6.62E-05 | 3.63E+04 | 7.2E+04 |
| $C_{10}H_{17}N_3O_{13}$ | LVOC | Yes | 1.33E-04 | 1.10E+04 | 7.2E+04 |
| $C_{10}H_{17}N_3O_{14}$ | ELVOC | Yes | 1.87E-04 | 7.96E+03 | 7.2E+04 |
| $C_{10}H_{17}N_3O_{15}$ | ELVOC | Yes | 3.61E-04 | 2.03E+03 | 7.2E+04 |
| $C_{10}H_{17}N_3O_{16}$ | ELVOC | Yes | 5.25E-04 | 1.10E+03 | 7.2E+04 |

[a] estimated by the median value of $C^*$ from different vapor pressure calculation methods used in this study

**References**

Aiken, A. C., Decarlo, P. F., Kroll, J. H., Worsnop, D. R., Huffman, J. A., Docherty, K. S., Ulbrich, I. M., Mohr, C., Kimmel, J. R., and Sueper, D.: O/C and OM/OC ratios of primary, secondary, and ambient organic aerosols with high-resolution time-of-flight aerosol mass spectrometry, Environ. Sci. Technol., 42, 4478-4485, 2008.

Al-Naiema, I. M., Hettiyadura, A. P., Wallace, H. W., Sanchez, N. P., Madler, C. J., Cevik, B. K., Bui, A. A., Kettler, J., Griffin, R. J., and Stone, E. A.: Source apportionment of fine particulate matter in Houston, Texas: insights to secondary organic aerosols, Atmos. Chem. Phys., 18, 15601-15622, 2018.

Bianchi, F., Kurten, T., Riva, M., Mohr, C., Rissanen, M. P., Roldin, P., Berndt, T., Crounse, J. D., Wennberg, P. O., Mentel, T. F., Wildt, J., Junninen, H., Jokinen, T., Kulmala, M., Worsnop, D. R., Thornton, J. A., Donahue, N., Kjaergaard, H. G., and Ehn, M.: Highly Oxygenated Organic Molecules (HOM) from Gas-Phase Autoxidation Involving Peroxy Radicals: A Key Contributor to Atmospheric Aerosol, Chem. Rev., 119, 3472-3509, 10.1021/acs.chemrev.8b00395, 2019.

Bilde, M., Barsanti, K., Booth, M., Cappa, C. D., Donahue, N. M., Emanuelsson, E. U., McFiggans, G., Krieger, U. K., Marcolli, C., Topping, D., Ziemann, P., Barley, M., Clegg, S., Dennis-Smither, B., Hallquist, M., Hallquist, A. M., Khlystov, A., Kulmala, M., Mogensen, D., Percival, C. J., Pope, F., Reid, J. P., Ribeiro da Silva, M. A., Rosenoern, T., Salo, K., Soonsin, V. P., Yli-Juuti, T., Prisle, N. L., Pagels, J., Rarey, J., Zardini, A. A., and Riipinen, I.: Saturation vapor pressures and transition enthalpies of low-volatility organic molecules of atmospheric relevance: from dicarboxylic acids to complex mixtures, Chem. Rev., 115, 4115-4156, 10.1021/cr5005502, 2015.

Brown, S., Degouw, J., Warneke, C., Ryerson, T., Dubé, W., Atlas, E., Weber, R., Peltier, R., Neuman, J., and Roberts, J.: Nocturnal isoprene oxidation over the Northeast United States in summer and its impact on reactive nitrogen partitioning and secondary organic aerosol, Atmos. Chem. Phys., 9, 3027-3042, 10.5194/acp-9-3027-2009, 2009.

Brown, S. S., and Stutz, J.: Nighttime radical observations and chemistry, Chem. Soc. Rev., 41, 6405-6447, 2012.

Brownwood, B., Turdziladze, A., Hohaus, T., Wu, R., Mentel, T. F., Carlsson, P. T., Tsiligiannis, E., Hallquist, M., Andres, S., and Hantschke, L.: Gas-particle partitioning and SOA yields of

organonitrate products from NO3-initiated oxidation of isoprene under varied chemical regimes, ACS Earth Space Chem., 2021.

Boyd, C., Sanchez, J., Xu, L., Eugene, A. J., Nah, T., Tuet, W., Guzman, M. I., and Ng, N.: Secondary organic aerosol formation from the β-pinene+ NO 3 system: effect of humidity and peroxy radical fate, Atmos. Chem. Phys., 15, 7497-7522, 2015.

Chen, Q., Farmer, D., Schneider, J., Zorn, S., Heald, C., Karl, T., Guenther, A., Allan, J., Robinson, N., and Coe, H.: Mass spectral characterization of submicron biogenic organic particles in the Amazon Basin, Geophys. Res. Lett., 36, 2009.

Chen, Y., Takeuchi, M., Nah, T., Xu, L., Canagaratna, M. R., Stark, H., Baumann, K., Canonaco, F., Prévôt, A. S., and Huey, L. G.: Chemical characterization of secondary organic aerosol at a rural site in the southeastern US: insights from simultaneous high-resolution time-of-flight aerosol mass spectrometer (HR-ToF-AMS) and FIGAERO chemical ionization mass spectrometer (CIMS) measurements, Atmos. Chem. Phys., 20, 8421-8440, 2020.

Chhabra, P., Flagan, R., and Seinfeld, J.: Elemental analysis of chamber organic aerosol using an aerodyne high-resolution aerosol mass spectrometer, Atmos. Chem. Phys., 10, 4111-4131, 2010.

Daellenbach, K. R., Kourtchev, I., Vogel, A. L., Bruns, E. A., Jiang, J., Petäjä, T., Jaffrezo, J.-L., Aksoyoglu, S., Kalberer, M., and Baltensperger, U.: Impact of anthropogenic and biogenic sources on the seasonal variation in the molecular composition of urban organic aerosols: a field and laboratory study using ultra-high-resolution mass spectrometry, Atmos. Chem. Phys., 19, 5973-5991, 2019.

Donahue, N. M., Epstein, S. A., Pandis, S. N., and Robinson, A. L.: A two-dimensional volatility basis set: 1. organic-aerosol mixing thermodynamics, Atmos. Chem. Phys., 11, 3303-3318, 10.5194/acp-11-3303-2011, 2011.

Fry, J. L., Brown, S. S., Middlebrook, A. M., Edwards, P. M., Campuzano-Jost, P., Day, D. A., Jimenez, J. L., Allen, H. M., Ryerson, T. B., Pollack, I., Graus, M., Warneke, C., de Gouw, J. A., Brock, C. A., Gilman, J., Lerner, B. M., Dubé, W. P., Liao, J., and Welti, A.: Secondary organic aerosol (SOA) yields from $NO_3$ radical + isoprene based on nighttime aircraft power plant plume transects, Atmos. Chem. Phys., 18, 11663-11682, 10.5194/acp-18-11663-2018, 2018.

Hu, W., Hu, M., Hu, W.-W., Zheng, J., Chen, C., Wu, Y., and Guo, S.: Seasonal variations in high time-resolved chemical compositions, sources, and evolution of atmospheric submicron aerosols in the megacity Beijing, Atmos. Chem. Phys., 17, 9979-10000, 2017.

Huang, X.-F., He, L.-Y., Hu, M., Canagaratna, M., Kroll, J., Ng, N., Zhang, Y.-H., Lin, Y., Xue, L., and Sun, T.-L.: Characterization of submicron aerosols at a rural site in Pearl River Delta of China using an Aerodyne High-Resolution Aerosol Mass Spectrometer, Atmos. Chem. Phys., 11, 1865-1877, 2011.

Kim, H., Zhang, Q., and Heo, J.: Influence of intense secondary aerosol formation and long-range transport on aerosol chemistry and properties in the Seoul Metropolitan Area during spring time: results from KORUS-AQ, Atmos. Chem. Phys., 18, 7149-7168, 2018.

Krechmer, J. E., Coggon, M. M., Massoli, P., Nguyen, T. B., Crounse, J. D., Hu, W., Day, D. A., Tyndall, G. S., Henze, D. K., and Rivera-Rios, J. C.: Formation of low volatility organic compounds and secondary organic aerosol from isoprene hydroxyhydroperoxide low-NO oxidation, Environ. Sci. Technol., 49, 10330-10339, 2015.

Kroll, J. H., Donahue, N. M., Jimenez, J. L., Kessler, S. H., Canagaratna, M. R., Wilson, K. R., Altieri, K. E., Mazzoleni, L. R., Wozniak, A. S., Bluhm, H., Mysak, E. R., Smith, J. D., Kolb, C. E., and Worsnop, D. R.: Carbon oxidation state as a metric for describing the chemistry of atmospheric organic aerosol, Nat. Chem., 3, 133-139, 10.1038/nchem.948, 2011.

Kurten, T., Tiusanen, K., Roldin, P., Rissanen, M., Luy, J.-N., Boy, M., Ehn, M., and Donahue, N.: α-Pinene autoxidation products may not have extremely low saturation vapor pressures despite high O: C ratios, J. Phys. Chem. A, 120, 2569-2582, 10.1021/acs.jpca.6b02196, 2016.

Lambe, A., Chhabra, P., Onasch, T., Brune, W., Hunter, J., Kroll, J., Cummings, M., Brogan, J., Parmar, Y., and Worsnop, D.: Effect of oxidant concentration, exposure time, and seed particles on secondary organic aerosol chemical composition and yield, Atmos. Chem. Phys., 15, 3063-3075, 2015.

Li, J., Cao, L., Gao, W., He, L., Yan, Y., He, Y., Pan, Y., Ji, D., Liu, Z., and Wang, Y.: Seasonal variations in the highly time-resolved aerosol composition, sources and chemical processes of background submicron particles in the North China Plain, Atmos. Chem. Phys., 21, 4521-4539, 2021.

Li, Y., Lee, B. P., Su, L., Fung, J. C. H., and Chan, C. K.: Seasonal characteristics of fine particulate matter (PM) based on high-resolution time-of-flight aerosol mass spectrometric (HR-ToF-

AMS) measurements at the HKUST Supersite in Hong Kong, Atmos. Chem. Phys., 15, 37-53, 2015.

Liu, J., Alexander, L., Fast, J. D., Lindenmaier, R., and Shilling, J. E.: Aerosol characteristics at the Southern Great Plains site during the HI-SCALE campaign, Atmos. Chem. Phys., 21, 5101-5116, 2021.

Massoli, P., Stark, H., Canagaratna, M. R., Krechmer, J. E., Xu, L., Ng, N. L., Mauldin, R. L., Yan, C., Kimmel, J., Misztal, P. K., Jimenez, J. L., Jayne, J. T., and Worsnop, D. R.: Ambient Measurements of Highly Oxidized Gas-Phase Molecules during the Southern Oxidant and Aerosol Study (SOAS) 2013, ACS Earth Space Chem., 2, 653-672, 10.1021/acsearthspacechem.8b00028, 2018.

Ng, N., Kwan, A., Surratt, J., Chan, A., Chhabra, P., Sorooshian, A., Pye, H. O., Crounse, J., Wennberg, P., and Flagan, R.: Secondary organic aerosol (SOA) formation from reaction of isoprene with nitrate radicals (NO3), Atmos. Chem. Phys., 8, 4117–4140, 10.5194/acp-8-4117-2008, 2008.

Ng, N. L., Brown, S. S., Archibald, A. T., Atlas, E., Cohen, R. C., Crowley, J. N., Day, D. A., Donahue, N. M., Fry, J. L., Fuchs, H., Griffin, R. J., Guzman, M. I., Herrmann, H., Hodzic, A., Iinuma, Y., Jimenez, J. L., Kiendler-Scharr, A., Lee, B. H., Luecken, D. J., Mao, J., McLaren, R., Mutzel, A., Osthoff, H. D., Ouyang, B., Picquet-Varrault, B., Platt, U., Pye, H. O. T., Rudich, Y., Schwantes, R. H., Shiraiwa, M., Stutz, J., Thornton, J. A., Tilgner, A., Williams, B. J., and Zaveri, R. A.: Nitrate radicals and biogenic volatile organic compounds: oxidation, mechanisms, and organic aerosol, Atmos. Chem. Phys., 17, 2103-2162, 10.5194/acp-17-2103-2017, 2017.

Nguyen, T. B., Bateman, A. P., Bones, D. L., Nizkorodov, S. A., Laskin, J., and Laskin, A.: High-resolution mass spectrometry analysis of secondary organic aerosol generated by ozonolysis of isoprene, Atmos. Environ., 44, 1032-1042, 2010.

Nguyen, T. B., Tyndall, G. S., Crounse, J. D., Teng, A. P., Bates, K. H., Schwantes, R. H., Coggon, M. M., Zhang, L., Feiner, P., and Milller, D. O.: Atmospheric fates of Criegee intermediates in the ozonolysis of isoprene, Phys. Chem. Chem. Phys., 18, 10241-10254, 2016.

Poulain, L., Spindler, G., Birmili, W., Plass-Dülmer, C., Wiedensohler, A., and Herrmann, H.: Seasonal and diurnal variations of particulate nitrate and organic matter at the IfT research station Melpitz, Atmos. Chem. Phys., 11, 12579-12599, 2011.

Schwantes, R. H., Teng, A. P., Nguyen, T. B., Coggon, M. M., Crounse, J. D., St Clair, J. M., Zhang, X., Schilling, K. A., Seinfeld, J. H., and Wennberg, P. O.: Isoprene NO3 Oxidation Products from the RO2 + HO2 Pathway, J. Phys. Chem. A, 119, 10158-10171, 10.1021/acs.jpca.5b06355, 2015.

Starn, T., Shepson, P., Bertman, S., Riemer, D., Zika, R., and Olszyna, K.: Nighttime isoprene chemistry at an urban‐impacted forest site, J. Geophys. Res. Atmos., 103, 22437-22447, 1998.

Stroud, C., Roberts, J., Williams, E., Hereid, D., Angevine, W., Fehsenfeld, F., Wisthaler, A., Hansel, A., Martinez‐Harder, M., and Harder, H.: Nighttime isoprene trends at an urban forested site during the 1999 Southern Oxidant Study, J. Geophys. Res. Atmos., 107, ACH 7-1-ACH 7-14, 2002.

Sun, Y., Zhang, Q., Macdonald, A., Hayden, K., Li, S., Liggio, J., Liu, P., Anlauf, K., Leaitch, W., and Steffen, A.: Size-resolved aerosol chemistry on Whistler Mountain, Canada with a high-resolution aerosol mass spectrometer during INTEX-B, Atmos. Chem. Phys., 9, 3095-3111, 2009.

Vereecken, L., Carlsson, P., Novelli, A., Bernard, F., Brown, S., Cho, C., Crowley, J., Fuchs, H., Mellouki, W., and Reimer, D.: Theoretical and experimental study of peroxy and alkoxy radicals in the NO 3-initiated oxidation of isoprene, Phys. Chem. Chem. Phys., 23, 5496-5515, 2021.

Wang, D. S., and Ruiz, L. H.: Secondary organic aerosol from chlorine-initiated oxidation of isoprene, Atmos. Chem. Phys., 17, 13491-13508, 2017.

Warneke, C., De Gouw, J., Goldan, P., Kuster, W., Williams, E., Lerner, B., Jakoubek, R., Brown, S., Stark, H., and Aldener, M.: Comparison of daytime and nighttime oxidation of biogenic and anthropogenic VOCs along the New England coast in summer during New England Air Quality Study 2002, J. Geophys. Res. Atmos., 109, 2004.

Wennberg, P. O., Bates, K. H., Crounse, J. D., Dodson, L. G., McVay, R. C., Mertens, L. A., Nguyen, T. B., Praske, E., Schwantes, R. H., and Smarte, M. D.: Gas-phase reactions of isoprene and its major oxidation products, Chem. Rev., 118, 3337-3390, 10.1021/acs.chemrev.7b00439, 2018.

Zhang, X., Xu, J., Kang, S., Zhang, Q., and Sun, J.: Chemical characterization and sources of submicron aerosols in the northeastern Qinghai–Tibet Plateau: insights from high-resolution mass spectrometry, Atmos. Chem. Phys., 19, 7897-7911, 2019.

Ziemann, P. J., and Atkinson, R.: Kinetics, products, and mechanisms of secondary organic aerosol formation, Chem. Soc. Rev., 41, 6582-6605, 10.1039/c2cs35122f, 2012.

Zhu, Q., He, L.-Y., Huang, X.-F., Cao, L.-M., Gong, Z.-H., Wang, C., Zhuang, X., and Hu, M.: Atmospheric aerosol compositions and sources at two national background sites in northern and southern China, Atmos. Chem. Phys., 16, 10283-10297, 2016.

---

## Author Comment (AC2) · 24 May 2021

**Response Letter to Referee #2**

The authors thank the reviewer for careful reading and positive feedbacks. We also appreciate the reviewer for the helpful comments and suggestions, which significantly improved the manuscript. All the comments are addressed below point by point in bold text, with our response followed in non-bold text and the corresponding revisions to the manuscript in blue. All updates to the original submission were tracked in the revised version as you can find below.

**Anonymous Referee #2:**

**The authors describe experimental results of the nitrate radical initiated oxidation of isoprene carried out in a batch-reactor for close to atmospheric conditions. Nitrate radical were generated via the O3+NO2 pathway. The reaction was conducted in such a way that multiple nitrate radical attacks were possible leading to product formation of the 2nd and 3rd generation. Product formation was followed by a bromide-CIMS. The results are interesting and worth to be published in ACP. Some points should be considered before acceptance of the manuscript can be recommended.**

**1. Line 71. "isoprene nitrates"? better: "organic nitrates derived from isoprene oxidation"**

**Response:** Accepted.

**2. Line 150, Fig. 1a-c: It is stated that approx. 90% of consumed isoprene reacted with NO3 based on modeling results. The reaction scheme with the used rate coefficients is not given yet. Please provide this information in the SI. The modeling results should be compared with the measurements of the chemical species depicted in Fig. 1 (as done in a couple of other**

**papers of the Jülich group). That gives the readership an impression how good the processes in the experiment have been understood and how accurate the model is.**

**Response:** Thanks very much for the suggestion, but there was a misunderstanding. The isoprene losses with respect to reaction with different oxidants were calculated from measurements, without using any modelling results. For example, the amount of isoprene consumed by $O_3$ was calculated from isoprene and $O_3$ measurements, and the isoprene consumed by $NO_3$ was calculated from the total isoprene losses using the measured isoprene and subtracting the losses due to $O_3$ and dilution. The reaction of isoprene with OH was not considered as its concentration was below the detection limit, and thus losses due to reaction with OH could not be quantified from the measurement. However, OH is expected to contribute about 10% of the isoprene losses, with the contribution of the $NO_3$ reaction accounting for up to 80%, as determined in a recently published modelling work based on the same campaign, which implemented a newly developed $NO_3$-isoprene mechanism with updated $RO_2$ and RO chemistry (Vereecken et al., 2021).

To avoid misleading, we revised the original sentence "According to the modeling results, approximately 90% of the isoprene reacted with $NO_3$, …" to "Calculation from measurements of isoprene, $O_3$, OH, $NO_3$ and dilution indicates that $NO_3$ contributed for more than 90% of the chemical losses of isoprene, as shown in Fig. S1, with reaction with $O_3$ being a minor pathway in our system. The reaction of isoprene with OH was not considered as OH concentration was below the detection limit of the instrument in this study (Fig. S2). Thus, losses due to reaction with OH could not be quantified from the measurement, but have been determined to contribute about 10% of the isoprene losses according to a recently published modelling work based on the same campaign, with the contribution of $NO_3$ reaction reducing to about 80% accordingly (Vereecken et al., 2021)."

[Figure]

**Figure S1: Fraction of isoprene consumed by O₃ and NO₃ calculated from measurements. The amount of isoprene consumed by O₃ was calculated from the measured concentrations of O₃ and isoprene measurements, and that consumed by NO₃ was calculated from total isoprene losses (from isoprene measurements) subtracting losses due to dilution and reaction with O₃. Isoprene losses due to reaction with OH could not be included here, because the OH concentration was below the detection limit. However, according the model calculations OH contributed about 10% to the isoprene consumption (Vereecken et al., 2021).**

**3. Line 200-212: The authors determined a bulk sensitivity for the organic nitrates. It should be described more in detailed what has been done. It is not clear to me why this calibration was not used to set all the measurements on an absolute scale. The authors argue "that the normalized signals are sufficient". More precise information is better in each case! On the other hand, they did it for C10 products. Why is the calibration only used for higher molecular products? Please comment. It would be fine if the authors could provide the plots of the C5 org. nitrate concentrations (or the yields because the amount of reacted isoprene is known). And please add a discussion regarding the uncertainty of these absolute values.**

**Response:** Sorry again for obviously unclear formulations.

To calculate the bulk sensitivity for organonitrates, the sum of organic nitrate signals from Br⁻ CIMS (in units of norm. count s⁻¹) was used and divided by measurements of the total alkyl

nitrates (in units of ppb) from a thermal dissociation-cavity ring-down spectrometer during the experiment. This gave us an idea about how large the concentrations may be at all.

In the revised manuscript, the original sentence in line 200-204 is revised to "Due to a lack of authentic standards for the products, it is difficult to quantitatively determine their individual absolute concentrations, but we calculated the bulk sensitivity for organonitrates using the sum of organic nitrate signals from Br⁻ CIMS divided by measurements of the total alkyl nitrates from a thermal dissociation-cavity ring-down spectrometer during the experiment."

The bulk sensitivity of Br⁻ CIMS for organic nitrates estimated in this study is a single-average value, so applying this value to convert the normalized signal to absolute concentration will not change any of the conclusions which are mostly based on the time behaviors of products (generations) and the relative changes of their normalized signals with time. Instead it may pretend a level of accuracy we cannot provide. By taking these two points into consideration, we prefer to keep using normalized signals rather than absolute concentrations for the analysis in this work.

However, as mentioned in the text, the bulk sensitivity was applied to estimate the potential SOA yield by condensation of the observed species. Here, several species contribute to SOA formation and application of an average sensitivity seems to be justified. The uncertainty of the estimated instrument sensitivity for organonitrates ($\sim 22\%$) is included in the calculations of the relative uncertainty of the SOA yield (32%) by error propagation, which includes also the uncertainty of the isoprene consumption concentration ($\sim 10\%$), the uncertainty of the alkyl nitrate concentration ($\sim 10\%$), and the uncertainty of the dimer content ($\sim 20\%$)., This is described in the caption of Fig. S8. In the revised manuscript, we add the uncertainty of the estimated SOA yield.

**4. Line 417, Fig. 2: In Fig.2 an average spectrum from the complete experiment is given. It would be fine having also a spectrum from the first injection showing mainly 1st generation products. A couple of possible reaction pathways were mentioned/ discussed in the paragraph before, incl. possible RO2 isomerization step leading possible to HOMs. Nothing is said here regarding the relevance of RO2 isomerization in this reaction system based on the measurements. It would be also very helpful for the readership to have a reaction scheme in the main body that explains the formation of the observed main products, i.e., C5H9NO5, C4H7NO5, C5H9NO4, C5H9NO6, … What about the formation of the carbonyls HCHO, MVK, MACR? The Vocus PTR-MS is very sensitive at least for the C4-carbonyls.**

**Response:** Thanks for the comment. Actually, mass spectra from different injections (denoted as step I, step II, and step IV) are shown in Fig. 3 in the manuscript (also shown below for illustration). We can determine from the plot that the major products in step I were 1N-monomers (in the mass range of *m/z* 220-280) and 2N-dimers (in the mass range of *m/z* 370-440), which are mostly expected to be first-generation products as discussed in the manuscript.

The general reaction schemes of $NO_3$-initiated oxidation of isoprene and the possible reaction pathways of the major products (including C5H9NOx, C4 carbonyls, C5 dinitrates and trinitrates, 2N- and 3N- dimers) are discussed in sect. 2.5.2, sect. 2.5.3 and sect. 2.5.4, with the corresponding schematic plots provided as supplementary materials. We moved on purpose the schemes to the supplement as they are not in the center of the manuscript but serve as a tool to get information about potential functionalization of the products for the vapor pressure calculations. We summarized all the underlying mechanistic information in Sect. 2.5, so that we did not need distribute them or repeat them throughout the text. Therefore, we did not repeat the discussion of the possible formation mechanisms of the major products in sect. 3. We would prefer to keep the manuscript as it is.

In the same sense, since this study focuses on the organic nitrate products of several generations which can be well detected by bromide CIMS, and we are more interested in multi-functional low-volatility molecules which can contribute to SOA formation, small compounds like HCHO, MVK/ MACR are of interest for the analysis in this study. Indeed, the referee is correct, Vocus identified C4 species with different chemical formulas than those measured by Br$^-$ CIMS. We only utilized the measurements of the isoprene precursor from Vocus. Analysis of the VOCUS data will surely provide many insights, but it is not needed for our purposes. A deeper analysis of VOCUS data is out of the scope of our manuscript.

[Figure]

**Figure 3: Comparison of the chemical composition of each oxidation step. (A) Averaged mass spectra for step I, II, and IV, with the omitted spectrum of step III being very similar to that of step II. (B) Relative contribution of different chemical groups for each oxidation step. Only organic products were counted for analysis. 'Others' refers to CHO compounds without containing nitrogen atoms (e.g., $C_5H_8O_2$ and $C_5H_8O_3$).**

**References**

Vereecken, L., Carlsson, P., Novelli, A., Bernard, F., Brown, S., Cho, C., Crowley, J., Fuchs, H., Mellouki, W., and Reimer, D.: Theoretical and experimental study of peroxy and alkoxy radicals in the NO3-initiated oxidation of isoprene, Phys. Chem. Chem. Phys., 23, 5496-5515, 2021.